# Decentralized Attention Fails Centralized Signals: Rethinking Transformers for Medical Time Series

**Guoqi Yu**[1], **Juncheng Wang**[1], **Chen Yang**[1], **Jing Qin**[2], **Angelica I. Aviles-Rivero**[3],
**Shujun Wang**[†1]
[1]Department of Biomedical Engineering, PolyU  [2]School of Nursing, PolyU
[3]Yau Mathematical Sciences Center, Tsinghua University

## Abstract

Accurate analysis of Medical time series (MedTS) data, such as Electroencephalography (EEG) and Electrocardiography (ECG), plays a pivotal role in healthcare applications, including the diagnosis of brain and heart diseases. MedTS data typically exhibits two critical patterns: **temporal dependencies** within individual channels and **channel dependencies** across multiple channels. While recent advances in deep learning have leveraged Transformer-based models to effectively capture temporal dependencies, they often struggle with modeling channel dependencies. This limitation stems from a structural mismatch: MedTS signals are inherently centralized, whereas the Transformer's attention is decentralized, making it less effective at capturing global synchronization and unified waveform patterns. To address this mismatch, we propose **CoTAR** (Core Token Aggregation-Redistribution), a centralized MLP-based module tailored to replace the decentralized attention. Instead of allowing all tokens to interact directly, as in attention, CoTAR introduces a global core token that acts as a proxy to facilitate the inter-token interaction, thereby enforcing a centralized aggregation and redistribution strategy. This design not only better aligns with the centralized nature of MedTS signals but also reduces computational complexity from quadratic to linear. Experiments on five benchmarks validate the superiority of our method in both effectiveness and efficiency, achieving up to a **11.6%** average-metric improvement on the APAVA dataset, with merely 33% memory usage and 20% inference time compared to the previous state-of-the-art. Code and all training scripts are available in `https://github.com/Levi-Ackman/TeCh`

## 1 Introduction

Medical time series (MedTS) data are temporal sequences of physiological data used to monitor a subject's health status (Badr et al., 2024), such as Electroencephalography (EEG) for neurological assessment (Arif et al., 2024; Jafari et al., 2023) and Electrocardiography (ECG) for cardiac diagnosis (Xiao et al., 2023; Wang et al., 2023). Accurate classification of MedTS facilitates early anomaly detection, timely diagnosis, and personalized treatment (Liu et al., 2021; Murat et al., 2020). This requires adequate modeling for two critical patterns: *temporal dependencies* within individual channels and *channel dependencies* across multiple channels, as illustrated in Figure 1 (a). Temporal dependencies reflect the intrinsic signal dynamics over time within each channel, such as oscillatory rhythms and event-related potentials for EEG (Niedermeyer & da Silva, 2005), and P&T wave for ECG (Goldberger et al., 2000). In contrast, channel dependencies capture the interactions and entanglements among multiple channels, such as functional connectivity for EEG (Stam, 2005) and the biophysical geometry of the heart for ECG (Macfarlane et al., 2005).

Previous deep-learning methods have achieved remarkable performance by focusing on modeling temporal dependencies, using architectures such as recurrent neural networks (RNNs) (Roy et al., 2019), convolutional neural networks (CNNs) (Wang et al., 2024a; Lawhern et al., 2018), or

---

[†] Correspondence to: Shujun Wang (e-mail: shu-jun.wang@polyu.edu.hk)

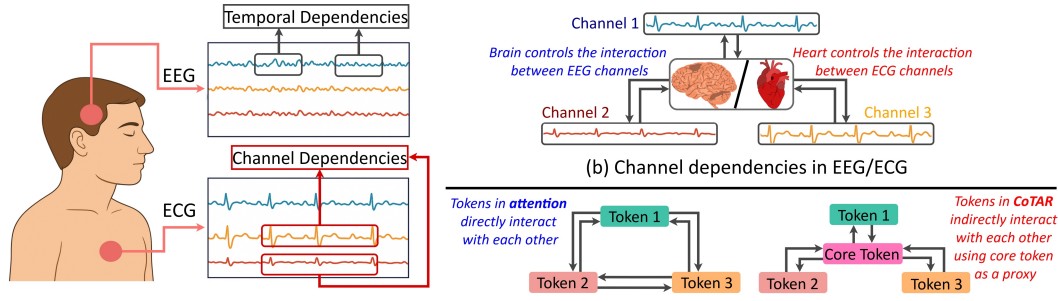

Figure 1: (a): Illustration of Temporal dependencies within each channel, and channel dependencies across channels. (b): Interaction between channels in EEG/ECG signals is centrally controlled by the brain/heart. (c): Attention module is a decentralized structure, where each token attends to all other tokens equally. (d): The proposed Core Token Aggregation-Redistribution (CoTAR) module operates in a centralized manner, with a core token as a proxy.

CNN–attention hybrids (Miltiadous et al., 2023a). However, each of these methods has limitations: RNNs suffer from sequential bottlenecks and difficulty capturing long-term dependencies, while CNNs are limited by local receptive fields and struggle with global temporal context. In contrast, Transformer (Vaswani et al., 2017) employs a decentralized attention mechanism, where each token can directly interact with all other tokens, which enables global receptive fields, allowing it to capture long-range and complex temporal dependencies effectively (Wang et al., 2025; Qiu et al., 2025; Liu et al., 2024a). This makes Transformer-based models deliver state-of-the-art MedTS classification performance (Wang et al., 2024b; Mobin et al., 2025). Despite their success in modeling temporal dependencies, Transformers face fundamental challenges when applied to modeling channel dependencies in MedTS. As illustrated in Figure 1 (b), **MedTS signals typically originate from a centralized biological source**. For example, EEG rhythms emerge from thalamo–cortical circuits synchronizing cortical neurons into coherent scalp oscillations (Schaul, 1998; Scherg et al., 2019), and ECG waveforms arise when impulses from the sinoatrial node propagate uniformly across the heart's conduction network (Rieta & Alcaraz, 1999; AlGhatrif & Lindsay, 2012). In contrast, Transformer's attention operates as a decentralized graph (Figure 1 (c)): every token attends equally to every other token (Vaswani et al., 2017; Gao et al., 2025). This uniform treatment of inter-channel interactions overlooks the inherent central coordination present in MedTS data. As a result, the attention mechanism tends to dilute the principal, centrally driven patterns, such as the cardiac pacemaker rhythms, and thus fails to capture the global synchronization and unified waveform features that are essential for accurate modeling of channel dependencies in MedTS.

To address this mismatch between the centralized nature of MedTS and the decentralized structure of attention, we ask: *can we maintain the benefits of attention (flexible, dynamic cross-channel interaction) while renovating it to reflect the centralized organization of MedTS?* Inspired by star-shaped architectures in distributed systems, where a central server mediates all communication for improved efficiency and robustness (Roberts & Wessler, 1970; Guo et al., 2019), we propose CoTAR (**Co**re **T**oken **A**ggregation-**R**edistribution): a lightweight, MLP-based module that seamlessly replaces the conventional attention. Instead of pairwise token interactions, CoTAR introduces a global core token that first aggregates information from all tokens and then redistributes it into each token, enabling centralized and flexible communication (Figure 1 (d)). This architecture not only better mirrors the central coordination inherent in signals like EEG and ECG, but also reduces the computational complexity of token interaction from **quadratic** to **linear**. This shift enables significant gains in scalability and efficiency, particularly for long or high-dimensional sequences common in medical applications (Arif et al., 2024; Jafari et al., 2023).

With CoTAR, we propose **TeCh**, a unified CoTAR-based framework that adaptively captures **Te**mporal dependencies, **Ch**annel dependencies, or both, by tuning the tokenization strategy (Temporal, Channel, or Dual). Such flexibility is particularly desirable in real-world medical time series, where not all datasets simultaneously exhibit strong temporal and inter-channel patterns. We conduct extensive experiments across five MedTS datasets, including three EEG datasets and two ECG datasets. Results show that Tech not only achieves the best performance across all datasets, but also introduces significantly lower resource consumption, highlighting its superior effectiveness, efficiency, and potential for broader real-world applications.

## 2 RELATED WORK

**Medical Time Series.** Medical time series (MedTS) are time series data collected from the human body, used for disease diagnosis (Liu et al., 2021; Xiao et al., 2023), health monitoring (Badr et al., 2024), and brain-computer interfaces (BCIs) (Musk et al., 2019; Altaheri et al., 2023). MedTS include EEG (Tang et al., 2021), ECG (Xiao et al., 2023), EMG (Xiong et al., 2021), and EOG (Jiao et al., 2020), each offering crucial information for medical applications. For example, EEG and ECG data are critical in assessing brain and heart health (Tang et al., 2021; Xiao et al., 2023). Such MedTS are characterized by temporal dependencies within each channel and channel dependencies between channels. Temporal dependencies include oscillatory rhythms and event-related potentials for EEG (Niedermeyer & da Silva, 2005), P wave and T wave for ECG (Goldberger et al., 2000). While the channel dependencies consist of functional connectivity for EEG (Stam, 2005), biophysical geometry of the heart for ECG (Macfarlane et al., 2005). Accurate modeling of these two patterns presents unique challenges. Recently, deep learning methods have significantly advanced the field of MedTS classification by providing precise temporal dependencies modeling using RNNs (Roy et al., 2019; Alhagry et al., 2017), CNNs (Lawhern et al., 2018), and Transformer (Wang et al., 2024b; Mobin et al., 2025), but the channel dependencies remain underexplored (Li et al., 2024; Fan et al., 2025; Kim et al., 2025).

**Transformers for Time Series.** Transformer-based models have been extensively adopted for time series analysis, with growing attention to both temporal and channel dependencies. For example, Informer (Zhou et al., 2021) proposes the Temporal embedding that aggregates values across channels as a token to model temporal dependencies. Autoformer (Wu et al., 2021) utilizes seasonal and trend decomposition to capture disentangled temporal information. PatchTST (Nie et al., 2023) splits the series from one channel into multiple patches, which improves the extraction of long-term temporal variations. iTransformer (Liu et al., 2024b) embeds the whole series of a channel into the Variate embedding, which maintains its complete context, thereby enhancing channel dependencies modeling. Finally, Leddam (Yu et al., 2024b) introduces a dual attention module to capture both temporal and channel dependencies.

Though the effective extraction of temporal dependencies has been addressed in MedTS using Temporal embedding and Transformer (Wang et al., 2024b; Mobin et al., 2025), the mismatch between the current decentralized attention structure and the centrally organized MedTS fails the Transformer in channel dependencies modeling. To address this, we propose a centralized MLP-based Core Token Aggregation-Redistribution (CoTAR) module, which delivers higher channel dependencies modeling ability while introducing only **Linear** complexity. By replacing attention using CoTAR, we propose a framework that can adaptively model **Te**mporal dependencies or **Ch**annel dependencies or both (denoted as **TeCh**) by tuning the tokenization strategy (Temporal, Channel, or Dual), whose effectiveness and efficiency are validated on five benchmarks.

## 3 PRELIMINARIES

**Subject-Independent Setting.** Medical time series (MedTS) data exhibit a hierarchical structure, spanning subjects (individuals), sessions (recordings per visit), trials (repeated measurements), and samples (short segments used for diagnosis model training) (Wang et al., 2024a). In clinical diagnosis tasks, the goal is to predict disease status at the subject level using tools such as deep models trained on MedTS samples. To ensure clinically meaningful evaluations, we adopt the '*Subject-Independent*' protocol (Wang et al., 2024c;b), which splits the dataset by subjects. Each subject, and all associated samples, appears exclusively in either the training, validation, or test set. This setting better reflects real-world deployment, where models must generalize to unseen patients, therefore providing a practical comparison.

**Problem Formulation.** *Consider an input MedTS sample $X \in \mathbb{R}^{T \times C}$, where $T$ denotes the number of timestamps and $C$ represents the number of channels. Our objective is to learn a function that can predict the corresponding label $\hat{Y} \in \mathbb{R}^{K}$. Here, $K$ denotes the number of classes, such as various disease types or different stages of one disease.*

## 4 METHOD

### 4.1 ATTENTION *vs* CORE TOKEN AGGREGATION-REDISTRIBUTION

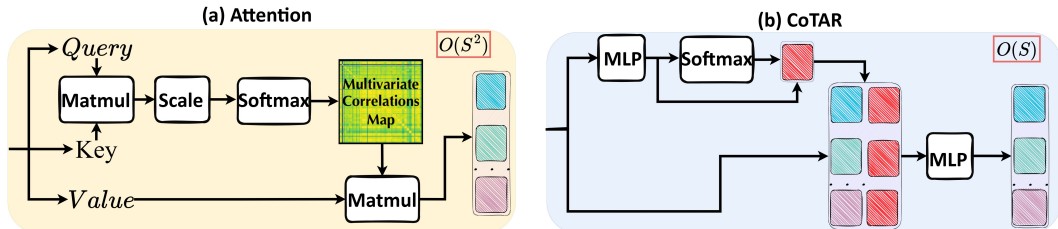

Figure 2: Illustration of attention and Core Token Aggregation-Redistribution (CoTAR). Attention is organized in a decentralized way where each token directly interacts with all tokens, introducing a **Quadratic** complexity. CoTAR first aggregates a core token and then redistributes it across channels to facilitate centralized channel interaction, bringing only **Linear** complexity.

**The standard Attention.** Transformer has demonstrated strong performance in many domains due to its ability to capture complex inter-token relationships, benefiting from the attention mechanism (Zhang et al., 2025a; 2024b; 2025b). Formally, for an input embedding $O \in \mathbb{R}^{S \times D}$ (where $S$ is the number of tokens and $D$ the embedding dimension), as in Figure 2 (a), attention operates via:

$$Q = OW_Q + b_q, \quad K = OW_K + b_k, \quad V = OW_V + b_v,$$

$$A = Softmax(\frac{QK^T}{\sqrt{D}})V, \quad Q, K, V, A \in \mathbb{R}^{S \times D}. \tag{1}$$

As mentioned before, such a decentralized structure does not fit the centrally controlled MedTS data. Besides, its quadratic complexity stemmed from the matrix multiplications between $Query$ and $Key$, making it inefficient for long and high-dimensional MedTS (Albuquerque et al., 2019).

**Core Token Aggregation-Redistribution (CoTAR).** To better match the MedTS and break the scalability bottleneck of attention, we borrow insight from the star-shaped centralized system in software engineering. Traditional peer-to-peer structure lets the clients communicate directly with each other, which is time- and resource-consuming. So a more reliable and efficient way is to set a server to aggregate and exchange the information between clients (Roberts & Wessler, 1970; Guo et al., 2019). Motivated by this, we propose the Core Token Aggregation-Redistribution (CoTAR), a plug-in module that can seamlessly replace attention, as shown in Figure 2 (b). CoTAR first projects the token of each channel, aggregates global context across channels into a core vector, and redistributes it back to every token. Given input $O \in \mathbb{R}^{S \times D}$, where $S$ denotes the number of tokens and $D$ the hidden dimension, CoTAR performs aggregation and redistribution as follows:

$$\begin{aligned}
\tilde{O} &= \text{GELU}(OW_1 + b_1)W_2 + b_2, & W_1 \in \mathbb{R}^{D \times D},\ b_1 \in \mathbb{R}^D,\ W_2 \in \mathbb{R}^{D \times D_c},\ b_2 \in \mathbb{R}^{D_c}, \\
O_w &= \text{Softmax}(\tilde{O}, \dim = 0), & \tilde{O} \in \mathbb{R}^{S \times D_c},\ O_w \in \mathbb{R}^{S \times D_c}, \\
\tilde{C}_o &= \text{Sum}(\tilde{O} \odot O_w, \dim = 0), & \tilde{C}_o \in \mathbb{R}^{D_c}, \\
C_o &= \text{Repeat}(\tilde{C}_o, \text{time} = S, \dim = 0), & C_o \in \mathbb{R}^{S \times D_c}, \\
O_{Co} &= \text{Concat}([O, C_o], \dim = 1), & O_{Co} \in \mathbb{R}^{S \times (D + D_c)}, \\
A &= \text{GELU}(O_{Co}W_3 + b_3)W_4 + b_4, & W_3 \in \mathbb{R}^{(D+D_c) \times D},\ W_4 \in \mathbb{R}^{D \times D},\ b_3, b_4 \in \mathbb{R}^D. \tag{2}
\end{aligned}$$

$D_c$ is the dimension of core token, $\tilde{C}_o$ **is the obtained core token** by aggregating information across all channels, and $A \in \mathbb{R}^{S \times D}$ is the final output. CoTAR employs a centralized structure that first gets the global core token by aggregating information from all channels. Then the core token is redistributed into each token. This realizes an indirect interaction between channels using the core token as a proxy (like the brain/heart in EEG/ECG). And since each token only needs to interact with a single core token, it only brings **Linear** complexity. Thus, CoTAR delivers higher effectiveness with lower resource consumption.

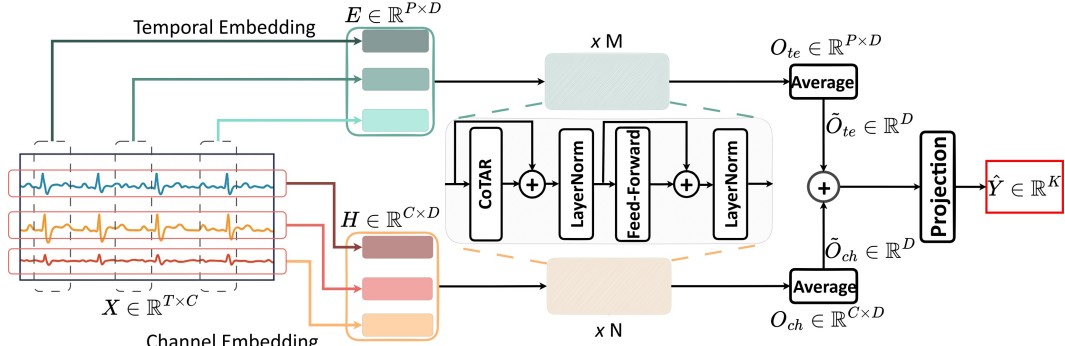

Figure 3: **Overview of TeCh.** MedTS signals $X \in \mathbb{R}^{T \times C}$ are embedded into Temporal embedding and Channel embedding. Then, each embedding is processed using Transformer encoders, with attention replaced by CoTAR. The final output representation from each branch is averaged across channels and added, then projected to the final predicted logits $\hat{Y} \in \mathbb{R}^{K}$.

## 4.2 OVERVIEW OF TeCh

The proposed Tech framework is illustrated in Figure 3. The raw MedTS is embedded into Temporal and Channel embedding, each is processed using a set of Transformer Encoders ($M$ for Temporal and $N$ for Channel, $M$ and $N$ are tunable to match with data, and the Temporal or Channel branch will be removed if $M = 0$ or $N = 0$); the learned representations are average across channels, fused and projected to the final output $\hat{Y} \in \mathbb{R}^{K}$.

**Adaptive Dual Tokenization.** Existing methods mainly rely on Temporal embedding that treats single or multiple timestamps across channels as a token, favoring temporal dependencies modeling while hindering channel dependencies extraction (Yu et al., 2024a; Qiu et al., 2024). So we take a balanced adaptive consideration of both patterns by using Adaptive Dual Tokenization.

Specifically, we form a temporal token by aggregating one or multiple timestamps across channels:

$$E_{i,:} = \text{vec}(X_{(i-1)L:iL,:})W_t + b_t + W_{i,:}^{tpos},$$
$$i = 1, \ldots, P, \quad P = \lceil T/L \rceil,$$
$$W_t \in \mathbb{R}^{LC \times D}, \quad b_t \in \mathbb{R}^{D}, \quad W^{tpos} \in \mathbb{R}^{P \times D}. \tag{3}$$

where $\text{vec} : \mathbb{R}^{m \times n} \to \mathbb{R}^{mn}$ flattens a 2D tensor into a 1D tensor, $L$ is a predefined hyperparameter that decides the granularity, $W^{tpos}$ is the classical position embedding (Vaswani et al., 2017). This will result in Temporal embedding $E \in \mathbb{R}^{P \times D}$. Then, following ***iTransformer*** (Liu et al., 2024b; Yu et al., 2025), we form a token by aggregating the whole series across all timestamps of a channel:

$$H_{j,:} = X_{:,j}^{\top}W_c + b_c + W_{j,:}^{cpos}, \quad j = 1, \ldots, C,$$
$$W_c \in \mathbb{R}^{T \times D}, \quad b_c \in \mathbb{R}^{D}, \quad W^{cpos} \in \mathbb{R}^{C \times D}. \tag{4}$$

This will result in Channel embedding $H \in \mathbb{R}^{C \times D}$. By embedding the whole series of each channel as a token, the unique semantic information of each individual channel is well-retained. Such a channel-centric token is proven to be effective in modeling multivariate correlations (Qiu et al., 2024; Wang et al., 2024d; Han et al., 2024).

In the real world, not all signals simultaneously exhibit strong temporal and inter-channel patterns. Thereby, our Adaptive Dual Tokenization strategy can better match with them by tuning $M$ and $N$.

**Classification Paradigm.** After Adaptive Dual Tokenization, the Temporal embedding $E$ and Channel embedding $H$ are processed using $M$ and $N$ standard Transformer Encoders with attention replaced by CoTAR, respectively. Then the learned Temporal representation $O_{te} \in \mathbb{R}^{P \times D}$ from the Temporal embedding is averaged across channels into $\tilde{O}_{te} \in \mathbb{R}^{D}$. Similarly, the learned Channel representation $O_{ch} \in \mathbb{R}^{C \times D}$ from the Channel embedding is averaged into $\tilde{O}_{ch} \in \mathbb{R}^{D}$. Notably,

if we set $M = 0$ or $N = 0$, this will remove the Temporal or Channel branch, and $\tilde{O}_{te} = 0$ or $\tilde{O}_{ch} = 0$. The final predicted logits are obtained via:

$$\hat{Y} = (\tilde{O}_{te} + \tilde{O}_{ch})W_y + b_y, \quad W_y \in \mathbb{R}^{D \times K}, \quad b_y \in \mathbb{R}^K. \quad (5)$$

With the Adaptive Dual Tokenization strategy, our Tech can adaptively model temporal dependencies or channel dependencies or both, and CoTAR allows for more effective and efficient token correlation extraction. These innovations make Tech a powerful, stable, and scalable framework for MedTS classification.

## 5 EXPERIMENTS

### 5.1 EXPERIMENT SETTING

We compare our **Tech** with 10 Transformer-based baselines across five MedTS datasets, including 3 EEG datasets, 2 ECG datasets. Our method is evaluated under the ***Subject-Independent*** setting, where training, validation, and test sets are split based on subjects. Additionally, we also conduct extensive experiments on two human activity recognition (HAR) datasets to test the generalizability.

Table 1: **The information of utilized datasets,** including the number of subjects, samples, classes, sample channels, and timestamps (TS).

| Dataset | #-Subject | #-Sample | #-Class | #-Channel | #-TS |
|---------|-----------|----------|---------|-----------|------|
| ADFTD   | 88        | 69,752   | 3       | 19        | 256  |
| APAVA   | 23        | 5,967    | 2       | 16        | 256  |
| TDBrain | 72        | 6,240    | 2       | 33        | 256  |
| PTB     | 198       | 64,356   | 2       | 15        | 300  |
| PTB-XL  | 17,596    | 191,400  | 5       | 12        | 250  |
| FLAPP   | 8         | 13123    | 10      | 6         | 100  |
| UCI-HAR | 30        | 10,299   | 6       | 9         | 128  |

**Datasets.** (1) **APAVA** (Escudero et al., 2006) is an EEG dataset where each sample is assigned a binary label indicating whether the subject has Alzheimer's disease. (2) **TDBrain** (van Dijk et al., 2022) is an EEG dataset with a binary label assigned to each sample, indicating whether the subject has Parkinson's disease. (3) **ADFTD** (Miltiadous et al., 2023b;a) is an EEG dataset with a three-class label for each sample, categorizing the subject as Healthy, having Frontotemporal Dementia, or Alzheimer's disease. (4) **PTB** (PhysioBank, 2000) is an ECG dataset where each sample is labeled with a binary indicator of Myocardial Infarction. (5) **PTB-XL** (Wagner et al., 2020) is an ECG dataset with a five-class label for each sample, representing various heart conditions. (6) **FLAAP** (Kumar & Suresh, 2022) is a smartphone-based HAR dataset that records accelerometer and gyroscope data for activity pattern recognition. (7) **UCI-HAR** (Anguita et al., 2013) comprises accelerometer and gyroscope data collected via waist-mounted smartphones, widely used for evaluating HAR models. Table 1 provides critical information, such as subjects, channels, and timestamps. The data preprocessing and dataset split follow Medformer (Wang et al., 2024b).

**Baselines.** We compare with 10 cutting-edge time series Transformer-based methods: Autoformer (Wu et al., 2021), FEDformer (Zhou et al., 2022), Informer (Zhou et al., 2021), iTransformer (Liu et al., 2024b), MTST (Zhang et al., 2024a), Nonformer (Liu et al., 2022), PatchTST (Nie et al., 2023), Reformer (Kitaev et al., 2019), vanilla Transformer (Vaswani et al., 2017), and Medformer (Wang et al., 2024b) (state-of-the-art Transformer-based MedTS classification model).

**Implementation.** We employ six evaluation metrics: accuracy, precision (macro-averaged), recall (macro-averaged), F1 score (macro-averaged), AUROC (macro-averaged), and AUPRC (macro-averaged). The training process is conducted with five random seeds (42-46) to compute the mean and standard deviation. All experiments are run on an NVIDIA RTX 4090 GPU. The results of all baselines on the five MedTS datasets are directly taken from Medformer (Wang et al., 2024b). And the results on the two HAR datasets are reproduced using the official code from Medformer (Wang et al., 2024b). We save the model with the best F1 score on the validation set.

Table 2: **Results on five MedTS datasets.** The training, validation, and test sets are distributed based on subject IDs. The best is **Bolded** and second is *Underlined*.

| Datasets | Models | Accuracy | Precision | Recall | F1 score | AUROC | AUPRC | Avg |
|---|---|---|---|---|---|---|---|---|
| ADFTD (3-Classes) | Autoformer | $45.25_{\pm1.48}$ | $43.67_{\pm1.94}$ | $42.96_{\pm2.03}$ | $42.59_{\pm1.85}$ | $61.02_{\pm1.82}$ | $43.10_{\pm2.30}$ | $46.43_{\pm1.90}$ |
| | FEDformer | $46.30_{\pm0.59}$ | $46.05_{\pm0.76}$ | $44.22_{\pm1.38}$ | $43.91_{\pm1.37}$ | $62.62_{\pm1.75}$ | $46.11_{\pm1.44}$ | $48.20_{\pm1.22}$ |
| | Informer | $48.45_{\pm1.96}$ | $46.54_{\pm1.68}$ | $46.06_{\pm1.84}$ | $45.74_{\pm1.38}$ | $65.87_{\pm1.27}$ | $47.60_{\pm1.30}$ | $50.04_{\pm1.57}$ |
| | iTransformer | $52.60_{\pm1.59}$ | $46.79_{\pm1.27}$ | $47.28_{\pm1.29}$ | $46.79_{\pm1.13}$ | $67.26_{\pm1.16}$ | $49.53_{\pm1.21}$ | $51.71_{\pm1.28}$ |
| | MTST | $45.60_{\pm2.03}$ | $44.70_{\pm1.33}$ | $45.05_{\pm1.30}$ | $44.31_{\pm1.74}$ | $62.50_{\pm0.81}$ | $45.16_{\pm0.85}$ | $47.89_{\pm1.34}$ |
| | Nonformer | $49.95_{\pm1.05}$ | $47.71_{\pm0.97}$ | $47.46_{\pm1.50}$ | $46.96_{\pm1.35}$ | $66.23_{\pm1.37}$ | $47.33_{\pm1.78}$ | $50.94_{\pm1.34}$ |
| | PatchTST | $44.37_{\pm0.95}$ | $42.40_{\pm1.13}$ | $42.06_{\pm1.48}$ | $41.97_{\pm1.37}$ | $60.08_{\pm1.50}$ | $42.49_{\pm1.79}$ | $45.56_{\pm1.37}$ |
| | Reformer | $50.78_{\pm1.17}$ | $49.64_{\pm1.49}$ | *$49.89_{\pm1.67}$* | $47.94_{\pm0.69}$ | *$69.17_{\pm1.58}$* | **$51.73_{\pm1.94}$** | $53.19_{\pm1.42}$ |
| | Transformer | $50.47_{\pm2.14}$ | $49.13_{\pm1.83}$ | $48.01_{\pm1.53}$ | $48.09_{\pm1.59}$ | $67.93_{\pm1.59}$ | $48.93_{\pm2.02}$ | $52.09_{\pm1.78}$ |
| | Medformer | *$53.27_{\pm1.54}$* | *$51.02_{\pm1.57}$* | **$50.71_{\pm1.55}$** | **$50.65_{\pm1.51}$** | **$70.93_{\pm1.19}$** | *$51.21_{\pm1.32}$* | **$54.63_{\pm1.45}$** |
| | TeCh | **$54.54_{\pm0.70}$** | **$53.02_{\pm0.87}$** | $49.25_{\pm1.01}$ | *$48.84_{\pm1.72}$* | $68.67_{\pm1.05}$ | $50.62_{\pm1.26}$ | *$54.16_{\pm1.10}$* |
| APAVA (2-Classes) | Autoformer | $68.64_{\pm1.82}$ | $68.48_{\pm2.10}$ | $68.77_{\pm2.27}$ | $68.06_{\pm1.94}$ | $75.94_{\pm3.61}$ | $74.38_{\pm4.05}$ | $70.71_{\pm2.63}$ |
| | FEDformer | $74.94_{\pm2.15}$ | $74.59_{\pm1.50}$ | $73.56_{\pm3.55}$ | $73.51_{\pm3.39}$ | $83.72_{\pm1.97}$ | $82.94_{\pm2.37}$ | $77.21_{\pm2.49}$ |
| | Informer | $73.11_{\pm4.40}$ | $75.17_{\pm6.06}$ | $69.17_{\pm4.56}$ | $69.47_{\pm5.06}$ | $70.46_{\pm4.91}$ | $70.75_{\pm5.27}$ | $71.36_{\pm5.04}$ |
| | iTransformer | $74.55_{\pm1.66}$ | $74.77_{\pm2.10}$ | $71.76_{\pm1.72}$ | $72.30_{\pm1.79}$ | *$85.59_{\pm1.55}$* | *$84.39_{\pm1.57}$* | $77.23_{\pm1.73}$ |
| | MTST | $71.14_{\pm1.59}$ | $79.30_{\pm0.97}$ | $65.27_{\pm2.28}$ | $64.01_{\pm3.16}$ | $68.87_{\pm2.34}$ | $71.06_{\pm1.60}$ | $69.94_{\pm1.99}$ |
| | Nonformer | $71.89_{\pm3.81}$ | $71.80_{\pm4.58}$ | $69.44_{\pm3.56}$ | $69.74_{\pm3.84}$ | $70.55_{\pm2.96}$ | $70.78_{\pm4.08}$ | $70.70_{\pm3.81}$ |
| | PatchTST | $67.03_{\pm1.65}$ | $78.76_{\pm1.28}$ | $59.91_{\pm2.02}$ | $55.97_{\pm3.10}$ | $65.65_{\pm0.28}$ | $67.99_{\pm0.76}$ | $65.89_{\pm1.52}$ |
| | Reformer | $78.70_{\pm2.00}$ | *$82.50_{\pm3.95}$* | $75.00_{\pm1.61}$ | $75.93_{\pm1.82}$ | $73.94_{\pm1.40}$ | $76.04_{\pm1.14}$ | $77.02_{\pm1.99}$ |
| | Transformer | $76.30_{\pm4.72}$ | $77.64_{\pm5.95}$ | $73.09_{\pm5.01}$ | $73.75_{\pm5.38}$ | $72.50_{\pm6.60}$ | $73.23_{\pm7.60}$ | $74.42_{\pm5.88}$ |
| | Medformer | *$78.74_{\pm0.64}$* | $81.11_{\pm0.84}$ | *$75.40_{\pm0.66}$* | *$76.31_{\pm0.71}$* | $83.20_{\pm0.91}$ | $83.66_{\pm0.92}$ | *$79.74_{\pm0.78}$* |
| | TeCh | **$86.86_{\pm1.09}$** | **$86.85_{\pm1.29}$** | **$86.10_{\pm1.00}$** | **$86.30_{\pm1.06}$** | **$94.02_{\pm0.52}$** | **$93.79_{\pm0.56}$** | **$88.99_{\pm0.92}$** |
| TDBrain (2-Classes) | Autoformer | $87.33_{\pm3.79}$ | $88.06_{\pm3.56}$ | $87.33_{\pm3.79}$ | $87.26_{\pm3.84}$ | $93.81_{\pm2.26}$ | $93.32_{\pm2.42}$ | $89.52_{\pm3.28}$ |
| | FEDformer | $78.13_{\pm1.98}$ | $78.52_{\pm1.91}$ | $78.13_{\pm1.98}$ | $78.04_{\pm2.01}$ | $86.56_{\pm1.86}$ | $86.48_{\pm1.99}$ | $80.98_{\pm1.96}$ |
| | Informer | $89.02_{\pm2.50}$ | $89.43_{\pm2.14}$ | $89.02_{\pm2.50}$ | $88.98_{\pm2.54}$ | $96.64_{\pm0.68}$ | $96.75_{\pm0.63}$ | $91.64_{\pm1.83}$ |
| | iTransformer | $74.67_{\pm1.06}$ | $74.71_{\pm1.06}$ | $74.67_{\pm1.06}$ | $74.65_{\pm1.06}$ | $83.37_{\pm1.14}$ | $83.73_{\pm1.27}$ | $77.63_{\pm1.11}$ |
| | MTST | $76.96_{\pm3.76}$ | $77.24_{\pm3.59}$ | $76.96_{\pm3.76}$ | $76.88_{\pm3.83}$ | $85.27_{\pm4.46}$ | $82.81_{\pm5.64}$ | $79.35_{\pm4.17}$ |
| | Nonformer | $87.88_{\pm2.48}$ | $88.86_{\pm1.84}$ | $87.88_{\pm2.48}$ | $87.78_{\pm2.56}$ | **$97.05_{\pm0.68}$** | **$96.99_{\pm0.68}$** | $91.07_{\pm1.79}$ |
| | PatchTST | $79.25_{\pm3.79}$ | $79.60_{\pm4.09}$ | $79.25_{\pm3.79}$ | $79.20_{\pm3.77}$ | $87.95_{\pm4.96}$ | $86.36_{\pm6.67}$ | $81.94_{\pm4.51}$ |
| | Reformer | $87.92_{\pm2.01}$ | $88.64_{\pm1.40}$ | $87.92_{\pm2.01}$ | $87.85_{\pm2.08}$ | $96.30_{\pm0.54}$ | $96.40_{\pm0.45}$ | $90.84_{\pm1.42}$ |
| | Transformer | $87.17_{\pm1.67}$ | $87.99_{\pm1.68}$ | $87.17_{\pm1.67}$ | $87.10_{\pm1.68}$ | $96.28_{\pm0.92}$ | $96.34_{\pm0.81}$ | $90.34_{\pm1.41}$ |
| | Medformer | *$89.62_{\pm0.81}$* | *$89.68_{\pm0.78}$* | *$89.62_{\pm0.81}$* | *$89.62_{\pm0.81}$* | $96.41_{\pm0.35}$ | $96.51_{\pm0.33}$ | *$91.91_{\pm0.65}$* |
| | TeCh | **$93.21_{\pm0.61}$** | **$93.39_{\pm0.58}$** | **$93.21_{\pm0.61}$** | **$93.20_{\pm0.61}$** | **$98.68_{\pm0.19}$** | **$98.72_{\pm0.17}$** | **$95.07_{\pm0.46}$** |
| PTB (2-Classes) | Autoformer | $73.35_{\pm2.10}$ | $72.11_{\pm2.89}$ | $63.24_{\pm3.17}$ | $63.69_{\pm3.84}$ | $78.54_{\pm3.48}$ | $74.25_{\pm3.53}$ | $70.86_{\pm3.17}$ |
| | FEDformer | $76.05_{\pm2.54}$ | $77.58_{\pm3.61}$ | $66.10_{\pm3.55}$ | $67.14_{\pm4.37}$ | $85.93_{\pm4.31}$ | $82.59_{\pm5.42}$ | $75.90_{\pm3.97}$ |
| | Informer | $78.69_{\pm1.68}$ | $82.87_{\pm1.02}$ | $69.19_{\pm2.90}$ | $70.84_{\pm3.47}$ | $92.09_{\pm0.53}$ | $90.02_{\pm0.60}$ | $80.62_{\pm1.70}$ |
| | iTransformer | *$83.89_{\pm0.71}$* | *$88.25_{\pm1.18}$* | $76.39_{\pm1.01}$ | $79.06_{\pm1.06}$ | $91.18_{\pm1.16}$ | *$90.93_{\pm0.98}$* | *$84.95_{\pm1.02}$* |
| | MTST | $76.59_{\pm1.90}$ | $79.88_{\pm1.90}$ | $66.31_{\pm2.95}$ | $67.38_{\pm3.71}$ | $86.86_{\pm2.75}$ | $83.75_{\pm2.84}$ | $76.80_{\pm2.68}$ |
| | Nonformer | $78.66_{\pm0.49}$ | $82.77_{\pm0.86}$ | $69.12_{\pm0.87}$ | $70.90_{\pm1.00}$ | $89.37_{\pm2.51}$ | $86.67_{\pm2.38}$ | $79.58_{\pm1.35}$ |
| | PatchTST | $74.74_{\pm1.62}$ | $76.94_{\pm1.51}$ | $63.89_{\pm2.71}$ | $64.36_{\pm3.38}$ | $88.79_{\pm0.91}$ | $83.39_{\pm0.96}$ | $75.35_{\pm1.85}$ |
| | Reformer | $77.96_{\pm2.13}$ | $81.72_{\pm1.61}$ | $68.20_{\pm3.35}$ | $69.65_{\pm3.88}$ | $91.13_{\pm0.74}$ | $88.42_{\pm1.30}$ | $79.51_{\pm2.17}$ |
| | Transformer | $77.37_{\pm1.02}$ | $81.84_{\pm0.66}$ | $67.14_{\pm1.80}$ | $68.47_{\pm2.19}$ | $90.08_{\pm1.76}$ | $87.22_{\pm1.68}$ | $78.69_{\pm1.52}$ |
| | Medformer | $83.50_{\pm2.01}$ | $85.19_{\pm0.94}$ | *$77.11_{\pm3.39}$* | *$79.18_{\pm3.31}$* | *$92.81_{\pm1.48}$* | $90.32_{\pm1.54}$ | $84.69_{\pm2.11}$ |
| | TeCh | **$85.96_{\pm2.52}$** | **$89.92_{\pm0.74}$** | **$79.43_{\pm4.13}$** | **$81.97_{\pm4.07}$** | **$94.57_{\pm0.70}$** | **$94.36_{\pm0.66}$** | **$87.70_{\pm2.14}$** |
| PTB-XL (5-Classes) | Autoformer | $61.68_{\pm2.72}$ | $51.60_{\pm1.64}$ | $49.10_{\pm1.52}$ | $48.85_{\pm2.27}$ | $82.04_{\pm1.44}$ | $51.93_{\pm1.71}$ | $57.53_{\pm1.88}$ |
| | FEDformer | $57.20_{\pm9.47}$ | $52.38_{\pm6.09}$ | $49.04_{\pm7.26}$ | $47.89_{\pm8.44}$ | $82.13_{\pm4.17}$ | $52.31_{\pm7.03}$ | $56.83_{\pm7.08}$ |
| | Informer | $71.43_{\pm0.32}$ | $62.64_{\pm0.60}$ | $59.12_{\pm0.47}$ | $60.44_{\pm0.43}$ | $88.65_{\pm0.09}$ | $64.76_{\pm0.17}$ | $67.84_{\pm0.35}$ |
| | iTransformer | $69.28_{\pm0.22}$ | $59.59_{\pm0.45}$ | $54.62_{\pm0.18}$ | $56.20_{\pm0.19}$ | $86.71_{\pm0.10}$ | $60.27_{\pm0.21}$ | $64.45_{\pm0.23}$ |
| | MTST | $72.14_{\pm0.27}$ | $63.84_{\pm0.72}$ | $60.01_{\pm0.81}$ | $61.43_{\pm0.38}$ | $88.97_{\pm0.33}$ | $65.83_{\pm0.51}$ | $68.70_{\pm0.50}$ |
| | Nonformer | $70.56_{\pm0.55}$ | $61.57_{\pm0.66}$ | $57.75_{\pm0.72}$ | $59.10_{\pm0.66}$ | $88.32_{\pm0.36}$ | $63.40_{\pm0.79}$ | $66.78_{\pm0.62}$ |
| | PatchTST | *$73.23_{\pm0.25}$* | *$65.70_{\pm0.64}$* | **$60.82_{\pm0.76}$** | **$62.61_{\pm0.34}$** | *$89.74_{\pm0.19}$* | **$67.32_{\pm0.22}$** | *$69.90_{\pm0.40}$* |
| | Reformer | $71.72_{\pm0.43}$ | $63.12_{\pm1.02}$ | $59.20_{\pm0.75}$ | $60.69_{\pm0.18}$ | $88.80_{\pm0.24}$ | $64.72_{\pm0.47}$ | $68.04_{\pm0.52}$ |
| | Transformer | $70.59_{\pm0.44}$ | $61.57_{\pm0.65}$ | $57.62_{\pm0.35}$ | $59.05_{\pm0.25}$ | $88.21_{\pm0.16}$ | $63.36_{\pm0.29}$ | $66.73_{\pm0.36}$ |
| | Medformer | $72.87_{\pm0.23}$ | $64.14_{\pm0.42}$ | $60.60_{\pm0.46}$ | $62.02_{\pm0.37}$ | $89.66_{\pm0.13}$ | $66.39_{\pm0.22}$ | $69.28_{\pm0.31}$ |
| | TeCh | **$73.53_{\pm0.07}$** | **$65.92_{\pm0.52}$** | *$60.61_{\pm0.59}$* | *$62.44_{\pm0.27}$* | **$90.03_{\pm0.12}$** | *$67.19_{\pm0.25}$* | **$69.95_{\pm0.30}$** |

## 5.2 MAIN RESULT

Table 2 presents the results under the Subject-Independent setup. Our TeCh consistently outperforms Medformer (the previous state-of-the-art) across all six metrics on four of the five MedTS datasets, achieving up to **11.6%** relative improvement in the ***average of all metrics*** on the APAVA dataset. Even in the challenging case of ADFTD, TeCh remains highly comparable to Medformer (Avg: 54.16 *vs.* 54.63) while ranking first in Accuracy and Precision and second in the overall Avg. Aggregated across all six metrics on these five MedTS datasets, TeCh achieves an overall **4.11%** relative performance gain over Medformer. In Table 3, TeCh substantially outperforms Medformer across all metrics on both datasets, with an average improvement of **4.28%**. Since HAR tasks involve multi-sensor channels and fine-grained activity classes, these consistent and significant gains indicate that TeCh generalizes better to noisy, high-variation, multi-channel time series inputs. In terms of robustness, TeCh also outperforms Medformer, as reflected in the lower average ***std*** across all datasets (0.84 *vs.* 0.96, a **12.37%** reduction). These results demonstrate that TeCh is both more effective and more robust than Medformer.

Table 3: **Results of two HAR datasets.** To evaluate the performance of our method on general time series, we test it on two human activity recognition (HAR) datasets: FLAAP and UCI-HAR, which exhibit potential channel correlations inherently. The best is **Bolded** and second is *Underlined*.

| Datasets | Models | Accuracy | Precision | Recall | F1 score | AUROC | AUPRC | Avg |
|---|---|---|---|---|---|---|---|---|
| **FLAAP** (10-Classes) | Autoformer | $38.93_{\pm1.01}$ | $38.22_{\pm1.31}$ | $37.40_{\pm1.17}$ | $33.51_{\pm1.14}$ | $74.12_{\pm0.35}$ | $35.77_{\pm0.91}$ | $42.99_{\pm0.98}$ |
| | FEDformer | $59.51_{\pm9.03}$ | $59.84_{\pm8.10}$ | $58.57_{\pm8.97}$ | $57.73_{\pm9.99}$ | $89.75_{\pm5.37}$ | $60.88_{\pm9.63}$ | $64.38_{\pm8.52}$ |
| | Informer | $72.87_{\pm0.89}$ | $73.20_{\pm0.97}$ | $72.76_{\pm0.92}$ | $72.59_{\pm0.96}$ | $95.91_{\pm0.24}$ | $77.57_{\pm1.21}$ | $77.48_{\pm0.87}$ |
| | iTransformer | $75.15_{\pm0.48}$ | $75.09_{\pm0.53}$ | $75.14_{\pm0.47}$ | $74.91_{\pm0.51}$ | $96.64_{\pm0.14}$ | $80.81_{\pm0.60}$ | $79.62_{\pm0.46}$ |
| | MTST | $70.57_{\pm0.54}$ | $71.09_{\pm0.73}$ | $70.97_{\pm0.73}$ | $70.61_{\pm0.57}$ | $94.56_{\pm0.18}$ | $73.28_{\pm0.99}$ | $75.18_{\pm0.62}$ |
| | Nonformer | $74.85_{\pm1.76}$ | $75.19_{\pm1.37}$ | $74.51_{\pm1.85}$ | $74.39_{\pm1.80}$ | $96.43_{\pm0.27}$ | $79.29_{\pm1.90}$ | $79.11_{\pm1.49}$ |
| | PatchTST | $56.34_{\pm0.31}$ | $56.36_{\pm0.63}$ | $55.29_{\pm0.32}$ | $55.58_{\pm0.45}$ | $89.24_{\pm0.11}$ | $58.92_{\pm0.36}$ | $61.96_{\pm0.36}$ |
| | Reformer | $71.13_{\pm1.64}$ | $71.20_{\pm1.81}$ | $70.57_{\pm1.66}$ | $70.54_{\pm1.79}$ | $95.16_{\pm0.42}$ | $73.80_{\pm2.09}$ | $75.40_{\pm1.57}$ |
| | Transformer | $76.36_{\pm1.21}$ | $76.53_{\pm1.25}$ | $76.23_{\pm0.98}$ | $76.05_{\pm1.16}$ | $96.65_{\pm0.11}$ | $80.70_{\pm0.63}$ | $80.42_{\pm0.89}$ |
| | Medformer | $76.44_{\pm0.64}$ | $76.61_{\pm1.13}$ | $76.63_{\pm1.36}$ | $76.25_{\pm0.65}$ | $95.44_{\pm0.26}$ | $81.12_{\pm1.60}$ | $80.42_{\pm0.94}$ |
| | TeCh | $\mathbf{80.60_{\pm0.30}}$ | $\mathbf{80.29_{\pm0.24}}$ | $\mathbf{80.36_{\pm0.32}}$ | $\mathbf{80.23_{\pm0.24}}$ | $\mathbf{97.67_{\pm0.10}}$ | $\mathbf{86.18_{\pm0.31}}$ | $\mathbf{84.22_{\pm0.25}}$ |
| **UCI-HAR** (6-Classes) | Autoformer | $41.86_{\pm2.46}$ | $49.62_{\pm11.48}$ | $44.30_{\pm2.55}$ | $32.69_{\pm2.60}$ | $83.72_{\pm2.53}$ | $58.56_{\pm4.67}$ | $51.79_{\pm4.38}$ |
| | FEDformer | $76.89_{\pm9.59}$ | $75.66_{\pm9.46}$ | $77.56_{\pm9.79}$ | $75.03_{\pm9.77}$ | $95.16_{\pm4.66}$ | $83.28_{\pm8.14}$ | $80.60_{\pm8.57}$ |
| | Informer | $88.33_{\pm1.26}$ | $88.28_{\pm1.20}$ | $88.47_{\pm1.20}$ | $88.20_{\pm1.29}$ | $98.36_{\pm0.14}$ | $94.20_{\pm0.33}$ | $90.97_{\pm0.90}$ |
| | iTransformer | $92.41_{\pm0.63}$ | $92.24_{\pm0.63}$ | $92.33_{\pm0.67}$ | $92.39_{\pm0.64}$ | $99.07_{\pm0.07}$ | $96.01_{\pm0.39}$ | $94.08_{\pm0.51}$ |
| | MTST | $90.99_{\pm0.84}$ | $90.96_{\pm0.79}$ | $90.92_{\pm0.85}$ | $90.83_{\pm0.88}$ | $98.21_{\pm0.11}$ | $96.14_{\pm0.59}$ | $93.01_{\pm0.68}$ |
| | Nonformer | $91.04_{\pm0.58}$ | $90.98_{\pm0.60}$ | $91.14_{\pm0.56}$ | $91.01_{\pm0.60}$ | $99.02_{\pm0.09}$ | $96.07_{\pm0.37}$ | $93.21_{\pm0.47}$ |
| | PatchTST | $87.67_{\pm0.39}$ | $88.37_{\pm0.43}$ | $87.97_{\pm0.37}$ | $88.02_{\pm0.38}$ | $98.50_{\pm0.09}$ | $93.86_{\pm0.40}$ | $90.73_{\pm0.34}$ |
| | Reformer | $88.70_{\pm1.14}$ | $88.82_{\pm1.03}$ | $88.82_{\pm1.13}$ | $88.59_{\pm1.19}$ | $98.68_{\pm0.26}$ | $94.60_{\pm1.07}$ | $91.37_{\pm0.97}$ |
| | Transformer | $89.36_{\pm1.74}$ | $89.33_{\pm1.70}$ | $89.49_{\pm1.69}$ | $89.33_{\pm1.75}$ | $98.87_{\pm0.23}$ | $95.58_{\pm0.68}$ | $91.99_{\pm1.30}$ |
| | Medformer | $89.62_{\pm0.81}$ | $89.70_{\pm0.18}$ | $89.80_{\pm0.14}$ | $89.62_{\pm0.81}$ | $98.11_{\pm0.06}$ | $94.80_{\pm0.72}$ | $91.94_{\pm0.45}$ |
| | TeCh | $\mathbf{94.15_{\pm0.96}}$ | $\mathbf{94.27_{\pm0.96}}$ | $\mathbf{94.30_{\pm0.97}}$ | $\mathbf{94.26_{\pm0.98}}$ | $\mathbf{99.32_{\pm0.05}}$ | $\mathbf{96.74_{\pm0.18}}$ | $\mathbf{95.51_{\pm0.68}}$ |

## 5.3 ABLATION STUDY

**Model Efficiency Analysis.** Since CoTAR introduces only **Linear** complexity compared to the **Quadratic** complexity of attention, our Tech achieves higher performance with significantly lower resource consumption, as in Figure 4 *(a)*. Compared to Medformer, Tech delivers **10.3%** better accuracy while using just 33% of the memory usage and 20% of the inference time.

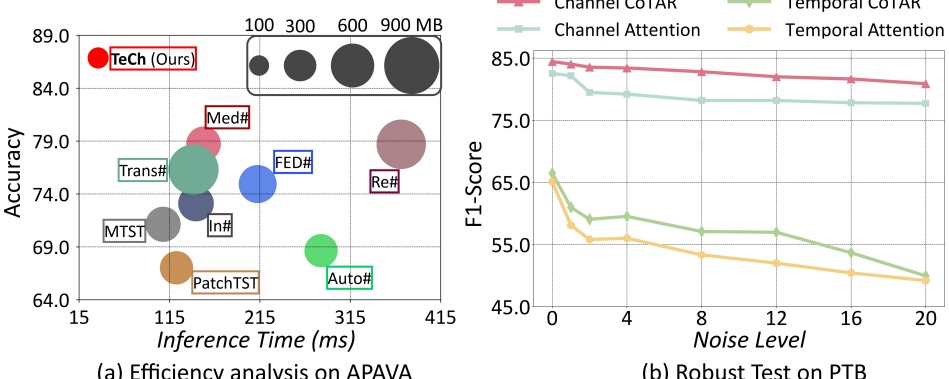

(a) Efficiency analysis on APAVA          (b) Robust Test on PTB

Figure 4: *(a)*: Efficiency and Effectiveness analysis of **TeCh** and other baselines on APAVA dataset with batch size $B = 128$. '#' stands for 'former' to save space. *(b)*: Robustness of attention and **CoTAR** to noise when using Channel or Temporal embedding. We consistently increase the intensity $\beta$ (the standard deviation) of Gaussian random noise from 0.0 to 20.0 on the last channel of the PTB dataset. F1-Score is used to quantify the change.

**Robustness Analysis.** To test the robustness of attention and CoTAR, we introduce noise progressively during training by adding perturbations to the last channel of the PTB dataset. This formulated as $\hat{X}_{:,C} = X_{:,C} + \beta \cdot \text{noise}$, where $X_{:,C}, \hat{X}_{:,C} \in \mathbb{R}^{1 \times T}$ is the last channel, $noise \in \mathbb{R}^{1 \times T}$ is Gaussian noise with mean 0 and standard deviation 1, $\beta \in \mathbb{R}^1$ controls the noise intensity. Then, the processed sample $\hat{X}_{:,C}$ is embedded into Channel embedding (Liu et al., 2024b) or Temporal embedding (Wang et al., 2024b). Figure 4 *(b)* reveals that attention is highly sensitive to noise. This is because attention is a decentralized structure, which means each channel can be directly influenced by the corrupted, noisy channel. In contrast, our CoTAR employed a centralized strategy, which prevents the noisy channel from directly interfering with others, therefore enhancing the robustness to noise. Meanwhile, compared to Temporal embedding, which is a more common practice in previous

Table 4: Ablation result of the proposed *Dual Tokenization* strategy. We include a general Human Activity dataset, UCI-HAR, to test its generalizability. *(i)* w/o: No tokenization is performed and directly uses the raw series as input-without representation learning, a single linear projection as classifier. *(ii)* Temporal: Only Temporal embedding is used. *(iii)* Channel: Only Channel embedding is used. *(iv)* Dual: Both Temporal and Channel embedding are used. The best is **Bolded**

| | ADFTD | | APAVA | | TDBrain | | PTB | | UCI-HAR | |
|---|---|---|---|---|---|---|---|---|---|---|
| | Accuracy | F1-Score | Accuracy | F1-Score | Accuracy | F1-Score | Accuracy | F1-Score | Accuracy | F1-Score |
| w/o | $33.79_{\pm0.64}$ | $32.67_{\pm0.53}$ | $50.68_{\pm0.86}$ | $50.13_{\pm0.88}$ | $53.79_{\pm1.21}$ | $53.77_{\pm1.20}$ | $72.62_{\pm1.30}$ | $64.84_{\pm2.05}$ | $54.22_{\pm0.47}$ | $51.72_{\pm0.47}$ |
| Temporal | $53.78_{\pm0.72}$ | $\mathbf{49.10_{\pm1.60}}$ | $55.93_{\pm5.06}$ | $53.71_{\pm5.56}$ | $\mathbf{93.21_{\pm0.61}}$ | $\mathbf{93.20_{\pm0.61}}$ | $74.74_{\pm0.55}$ | $62.90_{\pm1.15}$ | $91.56_{\pm0.63}$ | $91.52_{\pm0.62}$ |
| Channel | $47.06_{\pm1.35}$ | $32.92_{\pm0.90}$ | $75.68_{\pm1.80}$ | $73.54_{\pm2.49}$ | $67.58_{\pm1.04}$ | $67.54_{\pm1.06}$ | $\mathbf{85.96_{\pm2.52}}$ | $\mathbf{81.97_{\pm4.07}}$ | $92.98_{\pm0.44}$ | $93.00_{\pm0.48}$ |
| Both | $\mathbf{54.54_{\pm0.70}}$ | $48.84_{\pm1.72}$ | $\mathbf{86.86_{\pm1.09}}$ | $\mathbf{86.30_{\pm1.06}}$ | $89.79_{\pm0.96}$ | $89.77_{\pm0.97}$ | $84.15_{\pm2.06}$ | $79.11_{\pm3.43}$ | $\mathbf{94.15_{\pm0.96}}$ | $\mathbf{94.26_{\pm0.98}}$ |

Table 5: Ablation result of the proposed 'Core Token Aggregate-Redistribut' (CoTAR) module. *(i)* w/o: No Token interaction is performed, which means directly removing the CoTAR module. *(ii)* Attention: Replacing CoTAR with the Attention module. *(iii)* CoTAR: baseline with the CoTAR module. The best is **Bolded**.

| | ADFTD | | APAVA | | TDBrain | | PTB | | UCI-HAR | |
|---|---|---|---|---|---|---|---|---|---|---|
| | Accuracy | F1-Score | Accuracy | F1-Score | Accuracy | F1-Score | Accuracy | F1-Score | Accuracy | F1-Score |
| w/o | $53.32_{\pm0.67}$ | $47.26_{\pm0.53}$ | $83.31_{\pm0.95}$ | $81.99_{\pm1.18}$ | $92.69_{\pm0.75}$ | $92.67_{\pm0.76}$ | $85.28_{\pm2.32}$ | $80.82_{\pm3.69}$ | $92.40_{\pm0.19}$ | $92.55_{\pm0.21}$ |
| Attention | $52.77_{\pm1.00}$ | $48.65_{\pm1.22}$ | $83.42_{\pm1.60}$ | $82.09_{\pm0.28}$ | $90.40_{\pm2.18}$ | $90.35_{\pm2.23}$ | $85.74_{\pm1.45}$ | $81.93_{\pm2.22}$ | $93.13_{\pm0.59}$ | $93.21_{\pm0.60}$ |
| CoTAR | $\mathbf{54.54_{\pm0.70}}$ | $\mathbf{48.84_{\pm1.72}}$ | $\mathbf{86.86_{\pm1.09}}$ | $\mathbf{86.30_{\pm1.06}}$ | $\mathbf{93.21_{\pm0.61}}$ | $\mathbf{93.20_{\pm0.61}}$ | $\mathbf{85.96_{\pm2.52}}$ | $\mathbf{81.97_{\pm4.07}}$ | $\mathbf{94.15_{\pm0.96}}$ | $\mathbf{94.26_{\pm0.98}}$ |

work (Mobin et al., 2025; Wang et al., 2024b), Channel embedding delivers higher robustness and classification performance. This aligns with general time series analysis, where Channel embedding is more suitable for modeling channel dependencies, as it can better preserve the unique context of each channel, even when noise is entangled (Liu et al., 2024b; Wang et al., 2024d).

**Ablation Study on 'Adaptive Dual Tokenization'.** The results in Table 4 demonstrate the effectiveness of the proposed Adaptive Dual Tokenization design. When skipping the representation learning phase (the *w/o* setting), the performance significantly deteriorates across all datasets, highlighting the necessity of structured token embedding. Temporal tokenization excels on TDBrain, while Channel tokenization excels on PTB. And combining both yields an 11% improvement of Accuracy and a 13% improvement of F1-Score on APAVA. Moreover, Dual tokenization also excels on the UCI-HAR dataset, a well-known benchmark for Human Activity (HAR) tasks. Since HAR tasks involve multi-sensor channels and fine-grained activity classes, the significant gains of Dual Tokenization indicate that by simultaneously capturing both patterns, Tech can generalize to noisy, high-variation, multi-channel time series. These findings confirm that the Adaptive Dual Tokenization strategy enables Tech to better align with the unique characteristics of each dataset, providing more versatile modeling of Temporal dependencies or Channel dependencies, or both.

**Ablation Study on 'Core Token Aggregate-Redistribute'.** Table 5 provides a comprehensive ablation study validating the effectiveness of the proposed Core Token Aggregate-Redistribute (Co-TAR) module, which yields consistent performance gains across all five datasets and both metrics. Moreover, CoTAR also demonstrates competitive or lower standard deviations, indicating higher robustness. These results suggest that CoTAR not only captures richer inter-token dependencies through core-token centric redistribution but also leads to more stable and generalizable representations, thereby justifying its architectural necessity.

## 5.4 VISUALIZATION OF CORE TOKEN

In Figure 5, we visualized the *core token* generated by CoTAR and other embeddings across both temporal and channel spaces. Interestingly, in both embedding spaces, the core token consistently occupies a central position, suggesting that it captures a latent global physiological state integrating information across sensors (channel dimension) and across time (temporal dimension).

In the temporal space, this behavior reflects cross-temporal integration, which aggregates patterns over time into a stable representation of the system's evolving state. For EEG, such temporal aggregation resembles slow cortical dynamics, in which distributed neuronal populations maintain low-frequency coherence (*e.g.*, alpha or beta bands) to stabilize perception and working-memory states (Niedermeyer & da Silva, 2005; Buzsáki, 2006; Scherg et al., 2019). For ECG, it parallels the beat-to-beat coordination within the cardiac cycle: the sinus node's rhythmic discharge orchestrates each P–QRS–T sequence, and the consistent temporal integration of these cycles ensures stable and

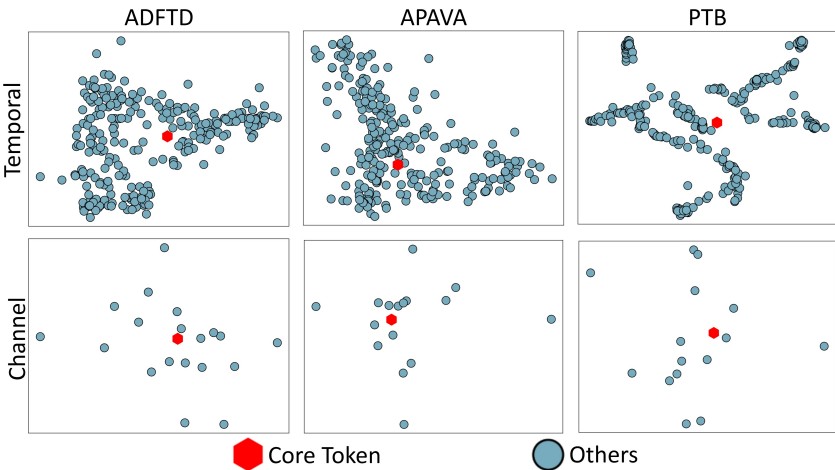

Figure 5: T-SNE visualization of the ***core token*** generated by CoTAR and other tokens. We visualize the embedding space of both temporal and channel.

regular cardiac pacing (AlGhatrif & Lindsay, 2012; Goldberger et al., 2000). Thus, the core token can be interpreted as a latent summary of temporal coherence in both neural and cardiac dynamics.

In the channel space, such centralization mirrors spatial integration across sensors. For EEG, this aligns with the global workspace and hub-based integration observed in frontoparietal networks that unify activity from distributed cortical regions (Dehaene & Changeux, 2011; Sporns, 2010). For ECG, it reflects pacemaker synchronization across myocardial conduction pathways, where a central excitation orchestrates coherent activation throughout the heart (Rieta & Alcaraz, 1999; AlGhatrif & Lindsay, 2012).

Together, these observations indicate that CoTAR's centralized proxy learns physiologically interpretable representations of both temporal and spatial coordination, effectively mirroring the centralized integration mechanisms that underlie real biological systems. We posit that such a centralized architecture could also inform the modeling of signals originating from centralized sources, *e.g.*, fMRI (Xun et al., 2025).

## 6    CONCLUSION

Existing Transformer models suffer from the mismatch between the centralized nature of medical time series (MedTS) and the decentralized structure of the attention module. This work proposes the Core Token Aggregation-Redistribution (**CoTAR**) module, which models inter-token relationships in a centralized way using a core token as a proxy, to replace attention seamlessly. Beyond being more effective in channel dependencies modeling, it also reduces the complexity from quadratic to linear. Based on CoTAR, our **TeCh** framework can adaptively capture temporal dependencies or channel dependencies, or both, and achieves superior performance and efficiency on three EEG and two ECG datasets, with a **4.11%** relative gain over Medformer. This work demonstrates the effectiveness of introducing domain-specific inductive biases into deep learning architectures for MedTS analysis and paves the way for more effective and scalable solutions.

## 7    ACKNOWLEDGEMENTS

This work was partially supported by the Research Grants Council (RGC) of Hong Kong under the Collaborative Research Fund (CRF) (No. C5055-24G), the Start-up Fund of The Hong Kong Polytechnic University (No. P0045999), the Seed Fund of the Research Institute for Smart Ageing (No. P0050946), the Tsinghua-PolyU Joint Research Initiative Fund (No. P0056509), and the University Grants Committee (UGC) funding of The Hong Kong Polytechnic University (No. P0053716).

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

# A    Data Augmentation Banks

In the embedding stage, we apply data augmentation to the input time series. We utilize a bank of data augmentation techniques to enhance the model's robustness and generalization. During the forward pass in training, each time series will pick one augmentation from available augmentation options with equal probability. The data augmentation methods include temporal flipping, channel shuffling, temporal masking, frequency masking, jittering, and dropout, and can be further expanded to include more choices. We provide a detailed description of each technique below.

**Temporal Flippling** We reverse the MedTS data along the temporal dimension. The probability of applying this augmentation is controlled by a parameter *prob*, with a default value of 0.5.

**Channel Shuffling** We randomly shuffle the order of MedTS channels. The probability of applying channel shuffling is controlled by the parameter *prob*, also set by default to 0.5.

**temporal masking** We randomly mask some timestamps across all channels. The proportion of timestamps masked is controlled by the parameter *ratio*, with a default value of 0.1.

**Frequency Masking** First introduced in (Zhang et al., 2022) for contrastive learning, this method involves converting the MedTS data into the frequency domain, randomly masking some frequency bands, and then converting it back. The proportion of frequency bands masked is controlled by the parameter *ratio*, with a default value of 0.1.

**Jittering** Random noise, ranging from 0 to 1, is added to the raw data. The intensity of the noise is adjusted by the parameter *scale*, which is set by default to 0.1.

**Dropout** Similar to the dropout layer in neural networks, this method randomly drops some values. The proportion of values dropped is controlled by the parameter *ratio*, with a default setting of 0.1.

# B    Data Preprocessing

We obtain all the well-preprocessed datasets from **Medformer** (Wang et al., 2024b) (`https://github.com/DL4mHealth/Medformer`). **Thanks for their brilliant work.**

## B.1    APAVA Preprocessing

The **A**lzheimer's **P**atients' Relatives **A**ssociation of **Va**lladolid (APAVA) dataset[1], referenced in the paper (Escudero et al., 2006), is a public EEG time series dataset with 2 classes and 23 subjects, including 12 Alzheimer's disease patients and 11 healthy control subjects. On average, each subject has $30.0 \pm 12.5$ trials, with each trial being a 5-second time sequence consisting of 1280 timestamps across 16 channels. Before further preprocessing, each trial is scaled using the standard scaler. Subsequently, we segment each trial into 9 half-overlapping samples, where each sample is a 1-second time sequence comprising 256 timestamps. This process results in 5,967 samples. Each sample has a subject ID to indicate its originating subject. For the training, validation, and test set splits, we employ the Subject-Independent setup. Samples with subject IDs {15,16,19,20} and {1,2,17,18} are assigned to the validation and test sets, respectively. The remaining samples are allocated to the training set.

## B.2    TDBrain Preprocessing

The TDBrain dataset[2], referenced in the paper (van Dijk et al., 2022), is a large permission-accessible EEG time series dataset recording brain activities of 1274 subjects with 33 channels. Each subject has two trials: one under eye open and one under eye closed setup. The dataset includes a total of 60 labels, with each subject potentially having multiple labels indicating multiple diseases simultaneously. In this paper, we utilize a subset of this dataset containing 25 subjects with Parkinson's disease and 25 healthy controls, all under the eye-closed task condition. Each eye-closed trial is segmented into non-overlapping 1-second samples with 256 timestamps, and any samples shorter

---

[1] `https://osf.io/jbysn/`
[2] `https://brainclinics.com/resources/`

than 1 second are discarded. This process results in 6,240 samples. Each sample is assigned a subject ID to indicate its originating subject. For the training, validation, and test set splits, we employ the Subject-Independent setup. Samples with subject IDs {18,19,20,21,46,47,48,49} are assigned to the validation set, while samples with subject IDs {22,23,24,25,50,51,52,53} are assigned to the test set. The remaining samples are allocated to the training set.

### B.3 ADFTD PREPROCESSING

The **A**lzheimer's **D**isease and **F**ron**T**otemporal **D**ementia (ADFTD) dataset[3], referenced in the papers (Miltiadous et al., 2023b;a), is a public EEG time series dataset with 3 classes, including 36 Alzheimer's disease (AD) patients, 23 Frontotemporal Dementia (FTD) patients, and 29 healthy control (HC) subjects. The dataset has 19 channels, and the raw sampling rate is 500Hz. Each subject has a trial, with trial durations of approximately 13.5 minutes for AD subjects (min=5.1, max=21.3), 12 minutes for FD subjects (min=7.9, max=16.9), and 13.8 minutes for HC subjects (min=12.5, max=16.5). A bandpass filter between 0.5-45Hz is applied to each trial. We downsample each trial to 256Hz and segment them into non-overlapping 1-second samples with 256 timestamps, discarding any samples shorter than 1 second. This process results in 69,752 samples. For the training, validation, and test set splits, we employ the Subject-Independent setup by allocating 60%, 20%, and 20% of total subjects with their corresponding samples into the training, validation, and test sets, respectively.

### B.4 PTB PREPROCESSING

The PTB dataset[4], referenced in the paper (PhysioBank, 2000), is a public ECG time series recording from 290 subjects, with 15 channels and a total of 8 labels representing 7 heart diseases and 1 health control. The raw sampling rate is 1000Hz. For this paper, we utilize a subset of 198 subjects, including patients with Myocardial infarction and healthy control subjects. We first downsample the sampling frequency to 250Hz and normalize the ECG signals using standard scalers. Subsequently, we process the data into single heartbeats through several steps. We identify the R-Peak intervals across all channels and remove any outliers. Each heartbeat is then sampled from its R-Peak position, and we ensure all samples have the same length by applying zero padding to shorter samples, with the maximum duration across all channels serving as the reference. This process results in 64,356 samples. For the training, validation, and test set splits, we employ the Subject-Independent setup. Specifically, we allocate 60%, 20%, and 20% of the total subjects, along with their corresponding samples, into the training, validation, and test sets, respectively.

### B.5 PTB-XL PREPROCESSING

The PTB-XL dataset[5], referenced in the paper (Wagner et al., 2020), is a large public ECG time series dataset recorded from 18,869 subjects, with 12 channels and 5 labels representing 4 heart diseases and 1 healthy control category. Each subject may have one or more trials. To ensure consistency, we discard subjects with varying diagnosis results across different trials, resulting in 17,596 subjects remaining. The raw trials consist of 10-second time intervals, with sampling frequencies of 100Hz and 500Hz versions. For our paper, we utilize the 500Hz version, then we downsample to 250Hz and normalize using standard scalers. Subsequently, each trial is segmented into non-overlapping 1-second samples with 250 timestamps, discarding any samples shorter than 1 second. This process results in 191,400 samples. For the training, validation, and test set splits, we employ the Subject-Independent setup. Specifically, we allocate 60%, 20%, and 20% of the total subjects, along with their corresponding samples, into the training, validation, and test sets, respectively.

---

[3]https://openneuro.org/datasets/ds004504/versions/1.0.6
[4]https://physionet.org/content/ptbdb/1.0.0/
[5]https://physionet.org/content/ptb-xl/1.0.3/

## C IMPLEMENTATION DETAILS

### C.1 IMPLEMENTATION DETAILS OF ALL BASELINES

We implement all the baselines based on the Medformer (Wang et al., 2024b), which integrates all methods under the same framework and training techniques to ensure a comprehensive, strict fair comparison. The compared 10 baseline time series transformer methods are Autoformer (Wu et al., 2021), FEDformer (Zhou et al., 2022), Informer (Zhou et al., 2021), iTransformer (Liu et al., 2024b), MTST (Zhang et al., 2024a), Nonformer (Liu et al., 2022), PatchTST (Nie et al., 2023), Reformer (Kitaev et al., 2019), Medformer (Wang et al., 2024b) (*the previous state-of-the-art*), and the vanilla Transformer (Vaswani et al., 2017).

For Medformer, we directly reproduced its result using their official implementations. For all other methods, we employ 6 layers for the encoder, with the self-attention dimension $D$ set to **128** and the hidden dimension of the feed-forward networks set to **256**. The optimizer used is Adam, with a learning rate of 1e-4. The batch size is set to {32,32,128,128,128} for the datasets APAVA, TDBrain, ADFD, PTB, and PTB-XL, respectively. Training is conducted for 100 epochs, with early stopping triggered after 10 epochs without improvement in the F1-Score on the validation set. We save the model with the best F1 score on the validation set and evaluate it on the test set. We employ six evaluation metrics: Accuracy, Precision (macro-averaged), Recall (macro-averaged), F1-Score (macro-averaged), AUROC (macro-averaged), and AUPRC (macro-averaged). Each experiment is run with 5 random seeds and fixed training, validation, and test sets to compute the average results and standard deviations.

**Autoformer** Autoformer (Wu et al., 2021) employs an auto-correlation mechanism to replace self-attention for time series forecasting. Additionally, they use a time series decomposition block to separate the time series into trend-cyclical and seasonal components for improved learning. The raw source code is available at `https://github.com/thuml/Autoformer`.

**FEDformer** FEDformer (Zhou et al., 2022) leverages frequency domain information using the Fourier transform. They introduce frequency-enhanced blocks and frequency-enhanced attention, which are computed in the frequency domain. A novel time series decomposition method replaces the layer norm module in the transformer architecture to improve learning. The raw code is available at `https://github.com/MAZiqing/FEDformer`.

**Informer** Informer (Zhou et al., 2021) is the first paper to employ a one-forward procedure instead of an autoregressive method in time series forecasting tasks. They introduce ProbSparse self-attention to reduce complexity and memory usage. The raw code is available at `https://github.com/zhouhaoyi/Informer2020`.

**iTransformer** iTransformer (Liu et al., 2024b) questions the conventional approach of embedding attention tokens in time series forecasting tasks and proposes an inverted approach by embedding the whole series of channels into a token. They also invert the dimension of other transformer modules, such as the layer norm and feed-forward networks. The raw code is available at `https://github.com/thuml/iTransformer`.

**MTST** MTST (Zhang et al., 2024a) uses the same token embedding method as Crossformer and PatchTST. It highlights the importance of different patching lengths in forecasting tasks and designs a method that can take different sizes of patch tokens as input simultaneously. The raw code is available at `https://github.com/networkslab/MTST`.

**Nonformer** Nonformer (Liu et al., 2022) analyzes the impact of non-stationarity in time series forecasting tasks and its significant effect on results. They design a de-stationary attention module and incorporate normalization and denormalization steps before and after training to alleviate the over-stationarization problem. The raw code is available at `https://github.com/thuml/Nonstationary_Transformers`.

**PatchTST** PatchTST (Nie et al., 2023) embeds a sequence of single-channel timestamps as a patch token to replace the attention token used in the vanilla transformer. This approach enlarges the receptive field and enhances forecasting ability. The raw code is available at `https://github.com/yuqinie98/PatchTST`.

**Reformer** Reformer (Kitaev et al., 2019) replaces dot-product attention with locality-sensitive hashing. They also use a reversible residual layer instead of standard residuals. The raw code is available at `https://github.com/lucidrains/reformer-pytorch`.

**Transformer** Transformer (Vaswani et al., 2017), commonly known as the vanilla transformer, was introduced in the well-known paper "Attention is All You Need." It can also be applied to time series by embedding each timestamp of all channels as an attention token. The PyTorch version of the code is available at `https://github.com/jadore801120/attention-is-all-you-need-pytorch`.

**Medformer** Medformer (Wang et al., 2024b) is a multi-granularity patching transformer specifically designed for medical time-series classification. It constructs patch tokens at multiple temporal resolutions to capture both fine-grained local dependencies and long-range contextual patterns. This design improves the model's ability to handle heterogeneous temporal dynamics in physiological signals. The raw code is available at `https://github.com/DL4mHealth/Medformer`.

## C.2 IMPLEMENTATION DETAILS OF OUR TECH

Our Tech is trained with a unified batch size ($B = 128$) and dimension of core token $D_c = \frac{1}{4}D$ across all datasets. The selection of other critical hyperparameters is listed in Table 6. We present the pseudo-code of the proposed CoTAR module in Algorithm 1.

Table 6: Critical hyperparameters for **TeCh** by dataset. We listed the model dimension ($D$), patch length of Temporal embedding ($L$), number of temporal encoders ($M$), number of channel encoders ($N$), and learning rate (**lr**).

| Dataset | $D$ | $L$ | $M$ | $N$ | lr |
|---------|-----|-----|-----|-----|------|
| ADFTD | 128 | 1 | 6 | 6 | 3e−5 |
| APAVA | 256 | 1 | 6 | 6 | 1e−4 |
| TDBRAIN | 128 | 6 | 6 | 0 | 1e−4 |
| PTB | 256 | 1 | 0 | 3 | 1e−4 |
| PTB-XL | 128 | 8 | 5 | 0 | 1e−4 |
| UCI-HAR | 256 | 12 | 5 | 6 | 1e−4 |
| FLAAP | 512 | 1 | 6 | 0 | 1e−4 |

---

**Algorithm 1** Pseudo-Code of Core Token Aggregation-Redistribution (CoTAR).

---

**Require: Input tensor:** $O \in \mathbb{R}^{S \times D}$.
**Require: Parameters:** Linear mapping layers Lin1, Lin2, Lin3, Lin4, dimension of core token $D_c$.
**Require: Definition:** $\text{Lin1} : \mathbb{R}^D \to \mathbb{R}^D$, $\text{Lin2} : \mathbb{R}^D \to \mathbb{R}^{D_c}$,
**Require: Definition:** $\text{Lin3} : \mathbb{R}^{D+D_c} \to \mathbb{R}^D$, $\text{Lin4} : \mathbb{R}^D \to \mathbb{R}^D$.
1: $\tilde{O} \leftarrow \text{Lin2}(\text{GELU}(\text{Lin1}(O)))$, $\tilde{O} \in \mathbb{R}^{S \times D_c}$,  ▷ First MLP to obtain core representation
2: $O_w \leftarrow \text{Softmax}(\tilde{O}, \dim = 0)$, $O_w \in \mathbb{R}^{S \times D_c}$,  ▷ Attention-like weights across channels
3: $\tilde{C}_o^{\ d} = \sum_{i=1}^{S} \tilde{O}^{i,d} \odot O_w^{i,d}$, $\tilde{C}_o \in \mathbb{R}^{D_c}$,  ▷ Weighted sum across channels to get core token
4: $C_o \leftarrow \text{Repeat}(\tilde{C}_o, N \text{ times})$, $C_o \in \mathbb{R}^{S \times D_c}$,  ▷ Repeat to align the channel dimension of input
5: $O_{Co} \leftarrow [O; C_o]$, $O_{Co} \in \mathbb{R}^{S \times (D+D_c)}$,  ▷ Concatenate along last dimension
6: $A \leftarrow \text{Lin4}(\text{GELU}(\text{Lin3}(O_{Co})))$, $A \in \mathbb{R}^{S \times D}$.  ▷ Fuse information through second MLP
7: **Return** $A \in \mathbb{R}^{S \times D}$

---

## C.3 FULL ABLATION RESULTS

To save space in the main text, we only present the ablation result of five representative datasets. We provide the full results on all datasets in Table 7 and Table 8.

Table 7: Full ablation result of the proposed ***Dual Tokenization*** strategy. *(i)* w/o: No tokenization is performed and directly uses the raw series as input-without representation learning, a single linear projection as classifier. *(ii)* Temporal: Only Temporal embedding. *(iii)* Channel: Only Channel embedding. *(iv)* Dual: Both Temporal and Channel. The best is **Bolded**

| | ADFTD | | APAVA | | TDBrain | | PTB | | PTB-XL | | FLAAP | | UCI-HAR | |
|---|---|---|---|---|---|---|---|---|---|---|---|---|---|---|
| | Accuracy | F1-Score | Accuracy | F1-Score | Accuracy | F1-Score | Accuracy | F1-Score | Accuracy | F1-Score | Accuracy | F1-Score | Accuracy | F1-Score |
| w/o | $33.79_{\pm0.64}$ | $32.67_{\pm0.53}$ | $50.68_{\pm0.86}$ | $50.13_{\pm0.88}$ | $53.79_{\pm1.21}$ | $53.77_{\pm1.20}$ | $72.62_{\pm1.30}$ | $64.84_{\pm2.05}$ | $30.95_{\pm0.13}$ | $20.61_{\pm0.51}$ | $28.54_{\pm2.34}$ | $25.08_{\pm1.33}$ | $54.22_{\pm0.47}$ | $51.72_{\pm0.47}$ |
| Temporal | $53.78_{\pm0.72}$ | $\mathbf{49.10_{\pm1.60}}$ | $55.93_{\pm5.06}$ | $53.71_{\pm5.56}$ | $\mathbf{93.21_{\pm0.61}}$ | $\mathbf{93.20_{\pm0.61}}$ | $74.74_{\pm0.55}$ | $62.90_{\pm1.15}$ | $\mathbf{73.53_{\pm0.07}}$ | $\mathbf{62.44_{\pm0.27}}$ | $\mathbf{80.60_{\pm0.30}}$ | $\mathbf{80.23_{\pm0.24}}$ | $91.56_{\pm0.63}$ | $91.52_{\pm0.62}$ |
| Channel | $47.06_{\pm1.35}$ | $32.92_{\pm0.90}$ | $75.68_{\pm1.80}$ | $73.54_{\pm2.49}$ | $67.58_{\pm1.04}$ | $67.54_{\pm1.06}$ | $\mathbf{85.96_{\pm2.52}}$ | $\mathbf{81.97_{\pm4.07}}$ | $69.18_{\pm0.21}$ | $54.76_{\pm0.47}$ | $77.48_{\pm0.13}$ | $77.06_{\pm0.17}$ | $92.98_{\pm0.44}$ | $93.00_{\pm0.48}$ |
| Both | $\mathbf{54.54_{\pm0.70}}$ | $48.84_{\pm1.72}$ | $\mathbf{86.86_{\pm1.09}}$ | $\mathbf{86.30_{\pm1.06}}$ | $89.79_{\pm0.96}$ | $89.77_{\pm0.97}$ | $84.15_{\pm2.06}$ | $79.11_{\pm3.43}$ | $73.15_{\pm0.09}$ | $62.13_{\pm0.16}$ | $78.03_{\pm0.31}$ | $77.86_{\pm0.30}$ | $\mathbf{94.15_{\pm0.96}}$ | $\mathbf{94.26_{\pm0.98}}$ |

Table 8: Full ablation result of the proposed 'Core Token Aggregate-Redistribut' (CoTAR) module. *(i)* w/o: No Token interaction is performed, which means directly removing the CoTAR module. *(ii)* Attention: Replacing CoTAR with the Attention module. *(iii)* CoTAR: baseline with the CoTAR module. The best is **Bolded**.

| | ADFTD | | APAVA | | TDBrain | | PTB | | PTB-XL | | FLAAP | | UCI-HAR | |
|---|---|---|---|---|---|---|---|---|---|---|---|---|---|---|
| | Accuracy | F1-Score | Accuracy | F1-Score | Accuracy | F1-Score | Accuracy | F1-Score | Accuracy | F1-Score | Accuracy | F1-Score | Accuracy | F1-Score |
| w/o | $53.32_{\pm0.67}$ | $47.26_{\pm0.53}$ | $83.31_{\pm0.95}$ | $81.99_{\pm1.18}$ | $92.69_{\pm0.75}$ | $92.67_{\pm0.76}$ | $85.28_{\pm2.32}$ | $80.82_{\pm3.69}$ | $72.25_{\pm0.38}$ | $59.48_{\pm0.59}$ | $74.48_{\pm0.46}$ | $74.00_{\pm0.53}$ | $92.40_{\pm0.19}$ | $92.55_{\pm0.21}$ |
| Attention | $52.77_{\pm1.00}$ | $48.65_{\pm1.22}$ | $83.42_{\pm1.60}$ | $82.09_{\pm0.28}$ | $90.40_{\pm2.18}$ | $90.35_{\pm2.23}$ | $85.74_{\pm1.45}$ | $81.93_{\pm2.22}$ | $72.01_{\pm0.22}$ | $60.96_{\pm0.21}$ | $77.16_{\pm0.76}$ | $76.87_{\pm0.77}$ | $93.13_{\pm0.59}$ | $93.21_{\pm0.60}$ |
| CoTAR | $\mathbf{54.54_{\pm0.70}}$ | $\mathbf{48.84_{\pm1.72}}$ | $\mathbf{86.86_{\pm1.09}}$ | $\mathbf{86.30_{\pm1.06}}$ | $\mathbf{93.21_{\pm0.61}}$ | $\mathbf{93.20_{\pm0.61}}$ | $\mathbf{85.96_{\pm2.52}}$ | $\mathbf{81.97_{\pm4.07}}$ | $\mathbf{73.53_{\pm0.07}}$ | $\mathbf{62.44_{\pm0.27}}$ | $\mathbf{80.60_{\pm0.30}}$ | $\mathbf{80.23_{\pm0.24}}$ | $\mathbf{94.15_{\pm0.96}}$ | $\mathbf{94.26_{\pm0.98}}$ |

## C.4 COMPARISON WITH CUTTING-EDGE TEMPORAL MODELS

To position TeCh within the broader landscape beyond current MedTS classifiers and relative to general time-series backbones exhibiting partial similarity, we present a comparative analysis that maps overlaps and distinctions between recent backbones and TeCh.

*(i) Methods employed a dual-dependencies modeling.* We select two representative works: GAFormer (ICLR24) (Xiao et al., 2024) and Leddam (ICML24) Yu et al. (2024b). GAFormer enhances token representations with group-aware embeddings for series clustering; Leddam introduces learnable decomposition into inter-series dependencies and intra-series variations; TeCh utilizes Adaptive Dual Tokenization (Temporal/Channel/Dual). Though all capture dual dependencies (temporal and inter-channel), GAFormer and Leddam target forecasting and are Transformer-based, thus decentralizing inter-channel interactions via attention, whereas TeCh uses a centralized CoTAR to better align with MedTS' biologically centralized sources (brain/heart). TeCh focuses on MedTS classification with physiological interpretability and linear complexity, while GAFormer/Leddam primarily focus on time series forecasting with quadratic attention costs. Consequently, GAFormer and Leddam are well-suited for broad forecasting scenarios; TeCh's centralized communication is more appropriate for MedTS channel dependencies. This is validated in our comparative result in Table 9. (Since there is no official implementation of GAFormer, and the information in the paper is not enough to reproduce, we take Leddam as baseline for its high reproducibility.)

*(ii) Methods employed global or auxiliary tokens.* We select two representative works: CATS (ICML24) (Lu et al., 2024) and TimeXer (NIPS24) (Wang et al., 2024e). They both employ global/auxiliary tokens that are parameter-initialized and learned jointly with the model, remaining largely input-agnostic while aggregating/redistributing information (often tied to exogenous-variable modeling). In contrast, TeCh's core token is generated adaptively from each input (subject) via CoTAR, making it data-conditional and thus better suited to MedTS heterogeneity where the "central source" differs across individuals. Moreover, TimeXer and CATS still operate within decentralized quadratic attention, while TeCh enforces centralized communication and achieves linear complexity. Additionally, TimeXer focuses on forecasting with exogenous variables and CATS constructs auxiliary time series to aid prediction, whereas TeCh targets MedTS classification with physiologically aligned central coordination. This dynamic, per-input core token mitigates the risk of poorer generalization from pre-defined global/aux tokens in clinical settings, as in Table 9. (Since there is no official implementation of CATS, and the information in the paper is not enough to reproduce, we take TimeXer as baseline for its high reproducibility.)

Table 9: We compare our **Tech** with two representative models in general time series analysis that are similar to ours in certain respects. *(i) **Leddam (Yu et al., 2024b)**:* like GAFormer (Xiao et al., 2024) and our Tech, all employ a dual-dependency modeling structure. *(ii) **TimeXer (Wang et al., 2024e)**:* like CATS (Lu et al., 2024) and our Tech, all employ global or auxiliary tokens to aggregate and redistribute information. The best is **Bolded**.

| | ADFTD | | APAVA | | TDBrain | | PTB | | PTB-XL | |
|---|---|---|---|---|---|---|---|---|---|---|
| | Accuracy | F1-Score | Accuracy | F1-Score | Accuracy | F1-Score | Accuracy | F1-Score | Accuracy | F1-Score |
| Leddam | $53.14_{\pm0.67}$ | $46.64_{\pm0.80}$ | $75.92_{\pm1.78}$ | $74.08_{\pm2.38}$ | $71.27_{\pm0.88}$ | $71.22_{\pm0.97}$ | $83.84_{\pm1.61}$ | $78.76_{\pm2.77}$ | $67.41_{\pm0.38}$ | $51.84_{\pm0.58}$ |
| TimeXer | $52.96_{\pm0.50}$ | $43.41_{\pm0.85}$ | $72.44_{\pm0.43}$ | $70.09_{\pm0.86}$ | $72.48_{\pm1.57}$ | $72.56_{\pm1.45}$ | $83.32_{\pm0.72}$ | $78.43_{\pm0.99}$ | $66.14_{\pm0.18}$ | $50.00_{\pm0.30}$ |
| Tech (Ours) | $\mathbf{54.54_{\pm0.70}}$ | $\mathbf{48.84_{\pm1.72}}$ | $\mathbf{86.86_{\pm1.09}}$ | $\mathbf{86.30_{\pm1.06}}$ | $\mathbf{93.21_{\pm0.61}}$ | $\mathbf{93.20_{\pm0.61}}$ | $\mathbf{85.96_{\pm2.52}}$ | $\mathbf{81.97_{\pm4.07}}$ | $\mathbf{73.53_{\pm0.07}}$ | $\mathbf{62.44_{\pm0.27}}$ |

Table 10: To further validate the generalizability, we further conduct a five-fold cross-validation based on the subject ID. The best is **Bolded**.

| | ADFTD | | APAVA | | TDBrain | | PTB | | PTB-XL | |
|---|---|---|---|---|---|---|---|---|---|---|
| | Accuracy | F1-Score | Accuracy | F1-Score | Accuracy | F1-Score | Accuracy | F1-Score | Accuracy | F1-Score |
| Medformer | $53.41_{\pm3.05}$ | $49.03_{\pm3.97}$ | $68.01_{\pm9.13}$ | $66.63_{\pm9.71}$ | $82.92_{\pm9.03}$ | $81.13_{\pm9.16}$ | $83.30_{\pm5.46}$ | $72.46_{\pm5.17}$ | $71.76_{\pm0.66}$ | $61.10_{\pm0.70}$ |
| Tech (Ours) | $\mathbf{55.05_{\pm2.43}}$ | $\mathbf{49.82_{\pm2.82}}$ | $\mathbf{80.66_{\pm6.53}}$ | $\mathbf{79.62_{\pm6.79}}$ | $\mathbf{87.06_{\pm6.71}}$ | $\mathbf{86.00_{\pm6.62}}$ | $\mathbf{89.48_{\pm3.18}}$ | $\mathbf{84.59_{\pm2.84}}$ | $\mathbf{73.65_{\pm0.41}}$ | $\mathbf{62.79_{\pm0.52}}$ |

Table 11: Quantitative comparison of the **centralized property**. We measure centralization using: (1) **Spectral Centralization Index (SCI)**, the ratio of the largest eigenvalue to total variance, and (2) **Dynamic Influence Centralization (DIC)**, the normalized out-strength imbalance of a first-order VAR model. Higher values indicate stronger centralized behavior.

| | EEG | | | ECG | | Energy | | Climate |
|---|---|---|---|---|---|---|---|---|
| Metric/Dataset | ADFTD | APAVA | TDBrain | PTB | PTB-XL | ETTh2 | ETTm2 | Weather |
| SCI | 0.918 | 0.520 | 0.616 | 0.622 | 0.652 | 0.397 | 0.296 | 0.381 |
| DIC | 0.668 | 0.731 | 0.747 | 0.825 | 0.777 | 0.241 | 0.119 | 0.342 |

Table 12: Further comparison with **MedGNN** (*the latest MedTS classifier in WWW 2025*).

| | ADFTD | | APAVA | | TDBrain | | PTB | | PTB-XL | |
|---|---|---|---|---|---|---|---|---|---|---|
| | Accuracy | F1-Score | Accuracy | F1-Score | Accuracy | F1-Score | Accuracy | F1-Score | Accuracy | F1-Score |
| MedGNN | $\mathbf{56.12_{\pm0.11}}$ | $\mathbf{55.00_{\pm0.24}}$ | $82.60_{\pm0.35}$ | $80.25_{\pm0.16}$ | $91.04_{\pm0.09}$ | $91.04_{\pm0.08}$ | $84.53_{\pm0.28}$ | $80.40_{\pm0.62}$ | $\mathbf{73.87_{\pm0.18}}$ | $\mathbf{62.54_{\pm0.20}}$ |
| Tech (Ours) | $54.54_{\pm0.70}$ | $48.84_{\pm1.72}$ | $\mathbf{86.86_{\pm1.09}}$ | $\mathbf{86.30_{\pm1.06}}$ | $\mathbf{93.21_{\pm0.61}}$ | $\mathbf{93.20_{\pm0.61}}$ | $\mathbf{85.96_{\pm2.52}}$ | $\mathbf{81.97_{\pm4.07}}$ | $73.53_{\pm0.07}$ | $62.44_{\pm0.27}$ |

## C.5 FIVE-FOLD CROSS-VALIDATION RESULT

To mitigate the bias of a fixed Subject-Independent split, we further performed a five-fold cross-validation based on subject IDs. As shown in Table 10, TeCh consistently surpasses Medformer across all datasets. For example, on APAVA, TeCh improves Accuracy and F1-Score by **+12.6%** and **+13.0%**, while on PTB, the gains reach **+6.2%** and **+12.1%**, respectively. TeCh also yields lower *standard deviation* (*e.g.*, *6.79 vs. 9.71* on APAVA F1-Score), indicating greater robustness. These results confirm that TeCh generalizes more effectively across subjects and remains robust to inter-subject noise, benefiting from CoTAR's centralized aggregating-redistributing mechanism.

## C.6 CENTRALIZATION ANALYSIS

To formally quantify the degree of centralization in a multivariate time series $\mathbf{J} \in \mathbb{R}^{S \times T}$, where $S$ is the number of channels, and $T$ is the length, we introduce two complementary metrics:

**(1) Spectral Centralization Index (SCI):**

$$\mathrm{SCI}(\mathbf{X}) = \frac{\lambda_{\max}\left(\frac{1}{T-1}(\mathbf{X} - \bar{\mathbf{X}})(\mathbf{X} - \bar{\mathbf{X}})^\top\right)}{\mathrm{Tr}\left(\frac{1}{T-1}(\mathbf{X} - \bar{\mathbf{X}})(\mathbf{X} - \bar{\mathbf{X}})^\top\right)}, \quad \bar{\mathbf{X}} = \frac{1}{T}\mathbf{X}\mathbf{1}_T.$$

**(2) Dynamic Influence Centralization (DIC):**

Let $Z = [\mathbf{x}_1, \mathbf{x}_2, \ldots, \mathbf{x}_{T-1}], \quad Y = [\mathbf{x}_2, \mathbf{x}_3, \ldots, \mathbf{x}_T],$

estimate $\mathbf{A} = Y Z^\dagger$, where $\mathbf{x}_t$ is the $t$-th column of $\mathbf{X}$.

$$\mathrm{DIC}(\mathbf{X}) = \frac{\max_i s_i - \bar{s}}{\bar{s}}, \quad \bar{s} = \frac{1}{S}\sum_i s_i, \quad s_i = \sum_j |A_{ji}|. \tag{6}$$

SCI measures spatial dominance as the energy concentration in the principal component of the co-variance matrix (Jolliffe & Cadima, 2016), while DIC captures temporal dominance as the normalized imbalance of out-strengths in a first-order vector autoregressive model (Seth et al., 2015; Valente et al., 2008). As shown in Table 11, EEG and ECG signals exhibit significantly higher centralization than signals generated from decentralized systems (Energy: ETTh2, ETTm2 (Zhou et al., 2021), Climate: Weather (Wu et al., 2021)). This confirms that MedTS possesses inherently centralized structures, where a few dominant channels or physiological processes govern the global dynamics. In contrast, energy and climate datasets are more decentralized. These findings explain why TeCh's centralized aggregating–redistributing design is particularly effective for MedTS.

## C.7 COMPARISON WITH THE LATEST ADVANCEMENT

We further compare against MedGNN, the latest MedTS classifier, in Table 12.

