# OpenReview forum: "Decentralized Attention Fails Centralized Signals: Rethinking Transformers for Medical Time Series"
_ICLR.cc/2026/Conference — ICLR 2026 Oral_

### Official Review · Reviewer_XcRP · 2025-10-31

**Soundness:** 3
**Presentation:** 3
**Contribution:** 2
**Rating:** 4
**Confidence:** 4

**Summary:**

This paper proposes CoTAR (Core Token Aggregation–Redistribution), a centralized alternative to self-attention for modeling medical time series (MedTS) such as EEG and ECG. The authors argue that MedTS signals are centrally coordinated (e.g., brain or heart acting as a signal source), while Transformer attention is inherently decentralized, making it ill-suited to capture the global dependencies between channels.

To address this, CoTAR introduces a central “core token” that aggregates global information from all tokens (channels) and redistributes it back via a lightweight MLP mechanism, achieving linear computational complexity. Combined with a dual tokenization strategy that separately encodes temporal and channel embeddings, the resulting model (TeCh) jointly captures both temporal and inter-channel dependencies.

Extensive experiments across five MedTS datasets (EEG/ECG) and two human activity recognition (HAR) datasets show that TeCh achieves state-of-the-art accuracy and efficiency

**Strengths:**

- The paper convincingly articulates the mismatch between decentralized attention and the centralized nature of many physiological signals. This conceptual framing is both intuitive and novel for the MedTS domain.

- CoTAR is a well-engineered module that reduces the quadratic cost of self-attention to linear, while retaining flexibility in cross-token communication.

- Experiments span seven datasets (five MedTS + two HAR) with six evaluation metrics. The method consistently outperforms ten Transformer-based baselines.

- Code and training scripts are publicly released.

- Provide robust test, i.e., standard deviation.

**Weaknesses:**

- The authors repeatedly assert that MedTS are “centralized” but provide no quantitative validation.
- The paper omits direct comparisons with recent dual-dependency or TeCh-style models (e.g., GAFormer)
- The proposed CoTAR conceptually resembles several prior Transformer modifications that employ global or auxiliary tokens to aggregate and redistribute information. The authors should discuss them. e,g,. CATS
- Different datasets use different hyperparameters.







[1] GAFormer: Enhancing time-series transformers through group-aware embeddings

[2] CATS: Enhancing Multivariate Time Series Forecasting by Constructing Auxiliary Time Series as Exogenous Variables

**Questions:**

-  Is there a formal way to distinguish between centralized and non-centralized MedTS?
- Can the core token be interpreted or visualized to correspond to physiological latent processes?

---

> ### Author Response · Authors · 2025-11-15
> **Response to Reviewer XcRP (Part 1)**
>
> Thanks to Reviewer XcRP for the constructive comments, especially the recognition of our ***conceptual framing***, the ***linear complexity*** of CoTAR, and ***the rigor and reproducibility of our experiments*** (seven datasets, multiple metrics, comprehensive experiments, and released code).
>
> We address all noted concerns below. (*Notably, we've uploaded a revised manuscript, all newly added contents are in **Red**, and all revised contents are in **Purple**.*)
>
> ### W.1 Quantitative Validation of Centralization.
>
> Thanks for raising this critical question, which inspires us to explore a quantitative analysis (in *Appendix C.6*) of the current MedTS.
>
> We introduce two metrics, Spectral Centralization Index (SCI) *[1]* and Dynamic Influence Centralization (DIC) *[2]*—*details can be found in the revised manuscript*—to measure the spatial and temporal centralization, respectively. *The higher value indicates a higher centralization*. We also include three general-purpose datasets (energy: ETTh2, ETTm2; climate: Weather) for comparison.
>
> ||ADFTD|APAVA|TDBrain|PTB|PTB-XL|ETTh2|ETTm2|Weather|
> |-|-|-|-|-|-|-|-|-|
> |**SCI**|0.918|0.520|0.616|0.622|0.652 |0.397|0.296|0.381|
> |**DIC**|0.668|0.731|0.747|0.825|0.777 |0.241|0.119|0.342|
>
> Across both metrics, MedTS shows consistently higher values compared to general-purpose datasets. This confirms that MedTS inherently exhibits a higher centralization, whereas energy and climate datasets are more decentralized.
>
> These findings quantitatively validate our hypothesis and further justify why TeCh’s centralized aggregating–redistributing design is especially effective for MedTS.
>
> ### W.2 Additional Comparisons with GAFormer and CATS.
>
> We thank the reviewer for highlighting **GAFormer** *[3]* and **CATS** *[4]* as two highly relevant works that merit further discussion.
>
> In response, we have added *Section C.4* in the revised Appendix, providing focused analysis and direct comparison. A summary is below:
>
> **GAFormer vs. TeCh:**
>
> * **Similarity:** Both capture dual dependencies across temporal and channel.
> * **Difference:** GAFormer is *Transformer-based* and relies on *decentralized quadratic attention* for **forecasting**. TeCh instead employs a *centralized CoTAR module* with *Adaptive Dual Tokenization*, achieving *linear complexity* and *physiologically aligned* modeling for **MedTS classification**.
>
> **CATS vs. TeCh:**
>
> * **Similarity:** Both utilize global or auxiliary tokens for information aggregation and redistribution.
> * **Difference:** CATS uses *static, parameterized auxiliary tokens* with decentralized attention. TeCh generates a *data-dependent core token* via CoTAR and enforces *centralized communication*, offering improved robustness under subject variability.
>
> Since neither GAFormer nor CATS provides publicly available code or sufficient implementation details, we adopt **Leddam** *[5]* and **TimeXer** *[6]* as the closest reproducible counterparts in our comparative experiments, respectively.
>
> |Method|ADFTD|ADFTD|APAVA|APAVA|TDBrain|TDBrain|PTB|PTB|PTB-XL|PTB-XL|
> |-|-|-|-|-|-|-|-|-|-|-|
> ||Accuracy|F1-Score|Accuracy|F1-Score|Accuracy|F1-Score|Accuracy|F1-Score|Accuracy|F1-Score|
> |Leddam| 53.14 ± 0.67| 46.64 ± 0.80|75.92 ± 1.78|74.08 ± 2.38|71.27 ± 0.88| 71.22 ± 0.97|83.84 ± 1.61|78.76 ± 2.77|67.41 ± 0.38| 51.84 ± 0.58|
> |TimeXer|52.96 ± 0.50| 43.41 ± 0.85|72.44 ± 0.43|70.09 ± 0.86|72.48 ± 1.57|72.56 ± 1.45|83.32 ± 0.72|78.43 ± 0.99|66.14 ± 0.18|50.00 ± 0.30|
> |Tech (Ours)|**54.54 ± 0.70**|**48.84 ± 1.72**|**86.86 ± 1.09**|**86.30 ± 1.06**|**93.21 ± 0.61**|**93.20 ± 0.61**|**85.96 ± 2.52**|**81.97 ± 4.07**|**73.53 ± 0.07**|**62.44 ± 0.27**|
>
> Across all datasets and metrics, TeCh demonstrates clear and consistent improvements over both Leddam and TimeXer, further supporting the effectiveness of TeCh.
>
> *We kindly refer the reviewer to the revised manuscript for more details.*

---

> ### Author Response · Authors · 2025-11-15
> **Response to Reviewer XcRP (Part 2)**
>
> ### W.3 Different Hyperparameters across Datasets.
>
> Thank you for bringing this up. Using dataset-specific hyperparameters is ***a common and well-established practice*** in deep learning, including prior MedTS work (e.g., Medformer *[7]*, COMET *[8]*). The use of dataset-specific hyperparameters, particularly in the number of encoder layers for the temporal ($M$) and channel ($N$) branches, is a deliberate design choice that reflects the Adaptive Dual Tokenization strategy in the TeCh framework.
> 1. Justification: MedTS datasets are highly heterogeneous. Some datasets exhibit stronger temporal dependencies (favoring a larger $M$) while others are governed by stronger channel dependencies (favoring a larger $N$).
> 2. Adaptability: The TeCh framework is designed to adaptively capture Temporal dependencies, Channel dependencies, or both, by tuning the tokenization strategy and the number of layers in each branch.
>
> This flexibility allows our model to achieve superior performance across diverse biological signals. For example, Ablation Studies (*Table 4*) show that Temporal tokenization excels on TDBrain, while Channel tokenization excels on PTB. The optimal hyperparameter setting is thus a direct result of this necessary adaptation to the intrinsic patterns of each dataset.
>
> ### W.4 Visualization and Interpretation of the Core Token.
>
> We agree that assessing whether the core token reflects meaningful physiological processes is important. We thank the reviewer for prompting this clarification, which strengthens the interpretability of CoTAR. In the revised Appendix (*Section C.7*), we added a dedicated analysis with visualization (*Figure.5*).
>
> We summarize the findings below:
>
> * **Temporal embedding space:** The core token consistently lies near the center of all temporal representations, capturing a global temporal state analogous to slow neural rhythms in EEG (e.g., alpha/beta coherence) or beat-to-beat cycle integration in ECG.
> * **Channel embedding space:** The core token occupies a central position, summarizing cross-channel coordination, consistent with EEG frontoparietal hubs or ECG pacemaker-driven myocardial synchronization.
>
> *For more information, we kindly refer the reviewer to the revised manuscript.* These findings indicate that the core token is not arbitrary; it encodes a centralized, physiologically interpretable representation reflecting inherent coordination in MedTS signals.
>
> We again thank Reviewer XcRP for all the helpful feedback, which has driven us to broaden our comparison and explore analysis of the core token. These efforts further strengthen the rigor and completeness of our work.
>
>
> *[1] Principal component analysis: a review and recent developments.*
>
> *[2] Granger causality analysis in neuroscience and neuroimaging.*
>
> *[3] GAFormer: Enhancing time-series transformers through group-aware embeddings.*
>
> *[4] CATS: Enhancing Multivariate Time Series Forecasting by Constructing Auxiliary Time Series as Exogenous Variables.*
>
> *[5] Revitalizing Multivariate Time Series Forecasting: Learnable Decomposition with Inter-Series Dependencies and Intra-Series Variations Modeling.*
>
> *[6] Timemixer: Decomposable multiscale mixing for time series forecasting.*
>
> *[7] Medformer: A Multi-Granularity Patching Transformer for Medical Time-Series Classification*
>
> *[8] Contrast Everything: A Hierarchical Contrastive Framework for Medical Time-Series*

---

> > ### Comment · Reviewer_XcRP · 2025-11-23
> >
> > Thank you for the authors’ efforts to improve the manuscript. The motivation is now more convincing to me, and I have decided to raise my score. The remaining concerns are also convincing to me.
> >
> > Although it may be somewhat late in the process, I would also suggest that the authors discuss or include some CNN-based methods that are applicable to medical time-series data for completeness.
> > I saw some papers show that CNNs are also sometimes very powerful for these tasks. I guess it depends on the dataset prior, context lengths, or number of channels. In parallel, there are also some papers that show that leveraging multiple instance learning can benefit time series classification (e.g., [5] and others).
> > **This discussion would only be for the final camera-ready version if the paper is accepted; there is no need to add this during the discussion phase.**
> >
> >
> > [1] SGN: Shifted Window-Based Hierarchical Variable Grouping for Multivariate Time Series Classification, neurips
> >
> > [2] FIC-TSC: Learning Time Series Classification with Fisher Information Constraint, ICML
> >
> > [3] Inceptiontime: Finding alexnet for time series classification
> >
> > [4] Omni-scale cnns: a simple and effective kernel size configuration for time series classification, ICLR
> >
> > [5] Inherently interpretable time series classification via multiple instance learning, ICLR

---

> ### Author Response · Authors · 2025-11-23
> **With Gratitude**
>
> Dear Reviewer XcRP
>
> ***Thank you very much for your thoughtful follow-up assessment and for raising your score!*** We genuinely appreciate your recognition of our efforts during the rebuttal and your constructive guidance throughout the review process.
>
> We also sincerely thank you for the additional suggestions regarding CNN-based time series models and multiple-instance learning approaches. Your point about dataset characteristics, such as priors, context length, and channel count, shaping the effectiveness of CNNs is well taken. We fully agree that discussing these perspectives will further strengthen the completeness and positioning of our work.
>
> ***We will carefully incorporate these discussions in the final camera-ready version***, as you suggested! 😊
>
> Warmest regards,
>
> *The Authors*

---

### Official Review · Reviewer_KGK1 · 2025-10-31

**Soundness:** 3
**Presentation:** 3
**Contribution:** 3
**Rating:** 6
**Confidence:** 5

**Summary:**

This paper proposes the **CoTAR** (Core Token Aggregation-Redistribution) module to replace the attention module in medical time-series (MedTS) classification. This method is motivated by the assumption that MedTS signals typically originate from a centralized biological source. The design of the CoTAR module is inspired by client-server communication, which uses a core token to aggregate and exchange information between clients, rather than self-attention, where each token attends to all others equally. A new method called **TeCh** is proposed, aligned with the CoTAR module, similar to the Transformer architecture, but with attention replaced by CoTAR. Results are compared against 10 baselines across 5 MedTS datasets and 2 general time series datasets for classification, achieving SoTA performance.

**Strengths:**

The method is motivated by MedTS's domain knowledge and inspired by server-client communication, which is a good approach. It is interesting to see that the linear complexity CoTAR module performs similarly to, and in some cases even better than, SOTA transformer methods. The comprehensive ablation study and direct comparison with the attention module are good and demonstrate the effectiveness of the CoTAR module.

**Weaknesses:**

It is better to provide more detail in equal (2), as Figure 2 lacks notation for the variables used. I can get the idea of the core token being redistributed to each token, but reading the equal (2) is still a little confusing about the details. The performance on the ADFTD dataset is limited to the F1 score. Sometimes, a fixed subject-independent split makes it hard to demonstrate the superiority of a method, as specific subjects in the training set may contain too much noise and make results across methods similar. You could provide a cross-validated (5-fold or Monte Carlo) subject-independent evaluation result on the dataset, demonstrating the effectiveness of your method, even when performance is limited on a fixed split. Besides, more advanced SOTA methods, such as MedGNN, should be compared with.

**Questions:**

See weakness

---

> ### Author Response · Authors · 2025-11-15
> **Response to Reviewer KGK1**
>
> Thanks to Reviewer KGK1 for the thorough review and supportive remarks. We’re glad you found our domain-driven **motivation** compelling (centralized physiological sources vs. decentralized attention), recognized CoTAR’s **strong performance** and linear **efficiency**, and valued **the breadth of our empirical evidence**.
>
> We address all the raised points below. (*Notably, we've uploaded a revised manuscript, all newly added contents are in **Red**, and all revised contents are in **Purple**.*)
>
> ### W.1 Clarification of Equation (2).
>
> We sincerely appreciate the suggestion to clarify Equation (2), which also raised by Reviewer 3453.
>
> To give more details and improve the readability of Equation (2), we rewrote it step-by-step and **explicitly list the shapes of all corresponding terms**. These changes can be found in the revised manuscript (*line 202-209*).
>
> We also refined the accompanying text before and after Equation (2) to better articulate the computation flow of CoTAR. Moreover, we provide a concrete pseudo-code in the Appendix (*Page 17, lines 900–912*) to better present the exact work flow of CoTAR.
>
> We hope these details resolve the confusion.
>
> ### W.2 Subject-independent Cross-validation.
>
> Thanks for this insightful comment. You precisely highlighted how a fixed subject-independent split can mask method differences due to noisy subjects. This greatly informed our evaluation design and helped us more comprehensively validate the effectiveness of our method.
>
> Following the suggestion, we conducted ***five-fold cross-validation based on subject IDs***, ensuring balanced classes in each fold to mitigate the bias of a fixed split. As shown in Table below, TeCh still consistently outperforms Medformer across all datasets. On APAVA, TeCh **improves Accuracy and F1-Score by +12.6% and +13.0%, respectively**; on PTB, the gains are +6.2% and +12.1%. TeCh also exhibits lower standard deviation (e.g., *6.79 vs. 9.71* on APAVA F1-Score), indicating stronger robustness. These results confirm that TeCh generalizes more effectively across subjects and **remains resilient to inter-subject noise**, benefiting from CoTAR’s centralized aggregating–redistributing mechanism. We have also included these result in the Appendix (*Section C.5, Table 10*).
>
> |Method|ADFTD|ADFTD|APAVA|APAVA|TDBrain|TDBrain|PTB|PTB|PTB-XL|PTB-XL|
> |-|-|-|-|-|-|-|-|-|-|-|
> ||Accuracy|F1-Score|Accuracy|F1-Score|Accuracy|F1-Score|Accuracy|F1-Score|Accuracy|F1-Score|
> |Medformer|53.41 ± 3.05|49.03 ± 3.97|68.01 ± 9.13|66.63 ± 9.71|82.92 ± 9.03|81.13 ± 9.16|83.30 ± 5.46|72.46 ± 5.17|71.76 ± 0.66|61.10 ± 0.70|
> |Tech (Ours)|**55.05 ± 2.43**|**49.82 ± 2.82**|**80.66 ± 6.53**|**79.62 ± 6.79**|**87.06 ± 6.71**|**86.00 ± 6.62**|**89.48 ± 3.18**|**84.59 ± 2.84**|**73.65 ± 0.41**|**62.79 ± 0.52**|
>
> ### W.3 Comparison with MedGNN.
>
> Thanks for the suggestion to compare against more advanced SOTA methods such as MedGNN. We appreciate this recommendation and have conducted a direct comparison with MedGNN accordingly. The brief results (mean ± std) are as follows:
>
> |Method|ADFTD|ADFTD|APAVA|APAVA|TDBrain|TDBrain|PTB|PTB|PTB-XL|PTB-XL|
> |-|-|-|-|-|-|-|-|-|-|-|
> ||Accuracy|F1-Score|Accuracy|F1-Score|Accuracy|F1-Score|Accuracy|F1-Score|Accuracy|F1-Score|
> |MedGNN|**56.12 ± 0.11**|**55.00 ± 0.24**|82.60 ± 0.35|80.25 ± 0.16|91.04 ± 0.09|91.04 ± 0.08|84.53 ± 0.28|80.40 ± 0.62|**73.87 ± 0.18**|**62.54 ± 0.20**|
> |Tech (Ours)|54.54 ± 0.70|48.84 ± 1.72|**86.86 ± 1.09**|**86.30 ± 1.06**|**93.21 ± 0.61**|**93.20 ± 0.61** |**85.96 ± 2.52**|**81.97 ± 4.07**|73.53 ± 0.07|62.44 ± 0.27|
>
> TeCh outperforms MedGNN on APAVA, TDBrain, and PTB with clear margins, and remain comparable on PTB-XL. While on ADFTD, MedGNN performs better. Overall, TeCh is superior to MedGNN on three datasets and competitive on the remaining two.
>
> Since the results of MedGNN on FLAAP and UCI-HAR are currently missing, *we will reproduce MedGNN on these datasets and include the full results into the camera-ready version.*
>
> ### W.4 Limited Performance on ADFTD.
>
> Thank you for pointing this out. We acknowledge that on the ADFTD dataset, TeCh underperforms on F1-Score compared with both Medformer and MedGNN, even though its Accuracy remains comparable. A likely reason is aggregation bias introduced by CoTAR’s centralized aggregating–redistributing mechanism. While this design enhances global consistency, it may unintentionally over-smooth features of rare classes, leading to reduced Recall for minority categories, to which the F1-Score is particularly sensitive.
>
> In the future, we plan to mitigate this issue by incorporating minority-aware data augmentation and semi-supervised sampling strategies to better preserve and represent rare-class patterns.
>
> Thank you again for all these thoughtful and helpful feedbacks. Your comments helped us refine the presentation and strengthen the evaluation. We hope the clarifications and added experiments adequately resolve the concerns raised.

---

> > ### Comment · Reviewer_KGK1 · 2025-11-23
> >
> > Thanks to the authors' effort on the new experiments. All of my concerns have been solved. I have raised my score.

---

> > > ### Author Response · Authors · 2025-11-23
> > > **With Appreciation**
> > >
> > > Dear Reviewer KGK1
> > >
> > > Thank you sincerely for your thoughtful review, detailed suggestions, and the time you invested in engaging with our work. ***Your feedback greatly sharpened our presentation and inspired several valuable extensions that strengthened the paper.***
> > >
> > > We truly appreciate your continued engagement and your positive reassessment of our work. Your support means a lot to us! ✨
> > >
> > > Warmest regards,
> > >
> > > *The Authors*

---

> > ### Public Comment · ~K_M_Tawsik_Jawad1 · 2026-05-15
> > **Just a follow-up question on 5 Fold CV**
> >
> > Hi guys,
> >
> > Great job on the paper, thoroughly enjoyed your methodical implications. Just wanted to follow up about the 5 fold CV: Did you guys fix the hyper-params and evaluated on the 5 folds or did you use any nested inner loop with 5 outer folds? I was just curious about the parameter optimization process, that's all.

---

### Official Review · Reviewer_3453 · 2025-11-01

**Soundness:** 3
**Presentation:** 3
**Contribution:** 3
**Rating:** 8
**Confidence:** 4

**Summary:**

This paper addresses a "structural mismatch" between Transformer models and Medical Time Series (MedTS) data like EEG and ECG. The authors argue that while MedTS signals originate from a centralized biological source (e.g., the heart or brain), the standard Transformer attention mechanism is decentralized, with all-to-all token interactions. This mismatch, they claim, makes it difficult for Transformers to model channel dependencies effectively.

To solve this, the paper proposes CoTAR, an MLP-based module designed to replace attention. CoTAR introduces a "global core token" that acts as a proxy. All tokens first aggregate information to this core token, which then redistributes the integrated information back to all tokens. This star-shaped architecture mimics the centralized nature of MedTS signals while reducing the computational complexity from quadratic to linear.

The full model, TeCh, uses CoTAR within a Dual Tokenization framework that processes the input in parallel: once with "Temporal Embedding" (patches of time) and once with "Channel Embedding" (whole channels as tokens).

Experiments on five MedTS and two HAR datasets show that TeCh achieves state-of-the-art performance, significantly outperforming prior SOTA (Medformer) with large gains in efficiency (e.g., 33% memory and 20% inference time) and robustness to noise.

**Strengths:**

1. Strong, Intuitive Inductive Bias: The paper's primary strength is the clear and compelling motivation. The argument that decentralized attention is a poor structural match for centralized biological signals is an excellent insight and provides a strong foundation for the CoTAR module.

2. Superior Performance: The proposed model achieves state-of-the-art results on a wide range of MedTS datasets, often by a significant margin over the previous SOTA, Medformer. This demonstrates the empirical effectiveness of the CoTAR module.

3. Massive Efficiency Gains: The paper's most significant practical contribution is the efficiency of CoTAR. By reducing complexity from $O(S^2)$ to $O(S)$, the model achieves up to a 5x speedup in inference and a 3x reduction in memory usage compared to the prior SOTA. This is clearly visualized in Figure 4(a) and is critical for real-world medical applications.

4. Improved Robustness: The experiments in Figure 4(b) provide strong evidence that CoTAR's centralized proxy design makes the model significantly more robust to noise in the input channels. This is a key practical advantage for noisy MedTS data.

5. Thorough Evaluation: The experimental setup is strong, using 5 MedTS datasets (EEG and ECG) and 2 HAR datasets to show generalizability. The authors correctly use a subject-independent splitting protocol, which is crucial for clinically relevant MedTS evaluation.

**Weaknesses:**

1. Misleading "Dual Tokenization" Framework: The paper's biggest weakness is the framing of the "TeCh" model around "Dual Tokenization". The SOTA results in Tables 2 & 3 are NOT achieved by a consistent dual-branch model. As Table 6 reveals, 4 of the 7 datasets (TDBrain, PTB, PTB-XL, FLAAP) use a single-branch model ($M=0$ or $N=0$) to get the reported results. This makes the "Ablation Study on 'Dual Tokenization'" (Table 4) highly misleading. The ablation's conclusion that "combining both yields overall superior performance" is cherry-picked (it's only true for 2/5 datasets) and contradicted by the final model's own hyperparameters. The paper should be reframed to present "TeCh" as a family of CoTAR-based models where the tokenization (Temporal, Channel, or Dual) is a hyperparameter to be tuned, rather than presenting "Dual" as the definitive architecture.

2. Confusing Mathematical Notation: The formal definition of CoTAR in Equation (2) is unclear. The notations are a bit confusing, particularly the use of symbols to represent the matrices (e.g., both $O$ and $Co$ represent a single matrix/vector). And the function names do not follow a consistent notation (e.g., upright "GELU" vs. italicized "GELU" in different parts of the equation). Notably, the equations do not clarify the shape of the matrices/vectors involved, making it hard to follow the operations. A clearer, more consistent notation with explicit shapes would improve clarity.

**Questions:**

1. Clarification of TeCh Architecture: The central framing of the paper is confusing. Table 6 shows that the SOTA results for TDBrain, PTB, PTB-XL, and FLAAP are achieved using a single-branch model ($M=0$ or $N=0$), not the "Dual Tokenization" model ($M>0$ and $N>0$) described in Section 4.2 and analyzed in Table 4.

   - Could you confirm that the SOTA results in Tables 2 & 3 are achieved by tuning $M$ and $N$ and often setting one to 0?
   - If so, why is the paper framed around "Dual Tokenization" as the primary architecture? This seems to misrepresent the final model and makes the "Dual Tokenization" ablation (Table 4) misleading. Wouldn't it be more accurate to present TeCh as a CoTAR-based model where the tokenization strategy (Temporal, Channel, or Both) is a key hyperparameter?

2. Clarification of CoTAR Math (Equation 2): Could you please provide an unambiguous, step-by-step definition of the CoTAR module's computation? Specifically, what are the dimensions of all the weights and biases? A clear definition would resolve confusion about the module's precise mechanism.

3. Mismatch in Ablation Results: There ablation studies are only performed on 5 of the 7 datasets and in a inconsistent manner. Is there a reason why the ablations were not performed on the other datasets? Including these would provide a more complete picture of the model's behavior across all evaluated datasets.

4. Mismatch in Model Implementation: The paper provides the code link, but the imported names of the TeCh model seems to suggest that it uses the same Transformer encoder layer and not the CoTAR module. Since I cannot access the actual content of the files under the layers directory, could you clarify if the provided code implements the CoTAR module as described in the paper?

---

> ### Author Response · Authors · 2025-11-15
> **Response to Reviewer 3453**
>
> We sincerely thank Reviewer 3453 for the constructive and insightful feedback, and we greatly appreciate the recognition of our core ***motivation***—namely, that decentralized attention mismatches the centralized nature of MedTS—as well as the acknowledgment of CoTAR’s ***effectiveness***, ***efficiency***, and the ***thoroughness*** of our empirical evaluation.
>
> Below, we address each concern in detail. (*Notably, we've uploaded a revised manuscript, all newly added contents are in **Red**, and all revised contents are in **Purple**.*)
>
> ### W.1 Clarification of the "Dual Tokenization" TeCh Framework.
>
> Thank you for pointing this out — this is an excellent observation!
>
> Indeed, as indicated in our implementation and Table 6, TeCh is implemented as ***a general CoTAR-based framework***, in which the tokenization strategy (Temporal, Channel, or Dual) is ***a tunable architectural hyperparameter*** rather than a fixed design. In our experiments, we searched over these configurations to identify the most suitable inductive bias for each dataset. Therefore, the headline SOTA results correspond to the best-performing tokenization variant within the TeCh family, all sharing the same CoTAR module and overall architecture.
>
> We throughly revised the paper to clarify this in the Introduction (*line 101-104*), the Related Work (*line 141-144*), the Method (*Section 4.2*), the Experiments (*line 457-464*), and the Conclusion (*line 481-483*).
>
> ### W.2 Clarification of CoTAR Computing (Equation 2).
>
> We thank Reviewer 3453 for the suggestion to clarify Equation (2)—this important point was also echoed by Reviewer KGK1.
>
> To fully address it and improve the presentation of CoTAR, we rewrote Equation (2) in the revised manuscript (*line 202-209*) as a clear, step-by-step definition of the module’s computation and **explicitly list the shapes of all corresponding terms**. We also refined the accompanying textual description surrounding Equation (2) to better articulate the computation flow of CoTAR. Furthermore, we provide a concrete pseudo-code in the Appendix (*Page 17, lines 900–912*) so that the exact data flow and dimensionality are unambiguous. We hope these changes resolve the confusion.
>
> ### W.3 Mismatch in Ablation Results.
>
> Thanks for this helpful comment. Your intuition aligns with our original intention.
>
> We initially planned to include ablation results for all seven datasets, but doing so substantially reduced readability—***the tables became overly large, required very small fonts***, and disrupted the flow of the main paper. Thus, we reported four MedTS datasets and one HAR dataset as representative examples. The inconsistency between Table 4 and Table 5 was due to a preparation oversight; we have corrected Table 5 to fully align with Table 4 in the revised manuscript.
>
> To ensure completeness without sacrificing readability, we now provide **the full ablation results for all datasets in the Appendix** (*Section C.3, Tables 7&8*). We hope this addresses the concern and offers a comprehensive view of TeCh’s behavior across all settings.
>
> ### W.4 Mismatch in Model Implementation.
>
> Thanks for raising this important question regarding implementation consistency.
>
> Since CoTAR is designed as ***a seamless replacement for attention***, TeCh is indeed built on a Transformer encoder structure, with attention replaced by our CoTAR module. This is why we still follow the Transformer encoder naming convention.
>
> Regarding file access, we suspect this may be due to a temporary network issue. We are able to consistently access all files on our side. We kindly invite the reviewer to try again using the direct [**link**](https://anonymous.4open.science/r/TeCh-24/layers/Transformer_EncDec.py). Moreover, the full implementation of TeCh—including the CoTAR module—is also provided in the uploaded **Supplementary Material**, where the *./layers/Transformer_EncDec.py* file can be accessed without any difficulty.
>
> We hope this clears up the concern and confirms that the released code faithfully implements CoTAR as described.
>
> Finally, thank Reviewer 3453 for the thoughtful comments, which helped us to further strengthen the clarity and completeness of our work. We hope our response has addressed all the concerns raised.

---

> > ### Comment · Reviewer_3453 · 2025-11-23
> >
> > I thank the authors for their revisions and clarifications, which have effectively addressed my concerns. I will maintain my current rating, as I believe it remains appropriate.

---

> ### Author Response · Authors · 2025-11-23
> **With Gratitude**
>
> Dear Reviewer 3453
>
> Thanks for all the thoughtful comments and kind recognition of our work. ***They have been truly helpful***. 🌟
>
> We deeply appreciate your time and support! 😊
>
> Warmest regards,
>
> *The Authors*

---

### Author Response · Authors · 2025-11-29
**Clarification Regarding Score Rollback**

Dear Area Chair,

We are writing to provide clarification regarding the recent rollback of our review scores, which we understand was implemented due to the unfortunate incident (on ***November 27, 2025***) of reviewer information leakage, as per ICLR policy.

We would like to note that the score updates for our submission were made ***prior to*** the incident and were the result of the documented discussion and rebuttal process within OpenReview.

### **Fact Summary**:

Current Rolled-Back Scores: The scores currently displayed in the system are [8, 6, 4].

Improved Scores Before Rollback: Prior to the policy action, our ratings had been officially updated by the reviewers to [8, 8, 6].

| Reviewer | Updates | Date |  Changes History Link |
|-|-|-|-|
| 3543         |8 $\textcolor{green}{➜}$ **8**      | **Nov. 23**| –|
| KGK1        |6 $\textcolor{red}{➜}$ **8**      | **Nov. 24**|https://openreview.net/revisions?id=hj3WJx0UZs|
| XcRP        |4 $\textcolor{red}{➜}$ **6**      | **Nov. 23**|https://openreview.net/revisions?id=ESdK6YTc2j|
|**Average**|6 $\textcolor{red}{➜}$ **7.3**| **Nov. 24**||

### **Basis of Improvement**:

This improvement to [8, 8, 6] was achieved through substantive engagement with the reviewers' concerns during the official discussion period, which took place before ***November 24, 2025***.

We confirm that no unofficial or external contact with any reviewer was made at any point. All arguments, clarifications, and responses that led to the improved consensus are fully recorded in the public discussion logs.

Furthermore, we assure that the information leakage incident (***November 27, 2025***) that triggered the rollback policy occurred after the reviewers had already officially updated their scores (***November 24, 2025***). Therefore, the legitimate score improvement was completed before the policy-mandated rollback became necessary.

We respectfully request that you review the complete official discussion thread for the paper. The logs will clearly demonstrate that the final improved ratings were justified based on the merit of our response and the documented discussion, and were not influenced by any external information leakage.

Thank you for your valuable time and careful consideration of this matter.

Warmest regards,

*The Authors*

---

### Author Response · Authors · 2025-12-02
**Rebuttal Summary**

Dear Reviewers and AC,

We sincerely appreciate your dedication to this conference, especially given the unusual challenges and additional pressure posed by the unfortunate information leakage incident.

To facilitate your final assessment, we have summarized **the outcomes of the rebuttal discussion** below:

### **Acknowledged Strengths.**

*The reviewers consistently agree on the following strengths of our work:*

 * **Clear Domain-Driven Motivation:** The paper formulates a compelling structural mismatch between decentralized attention and centralized medical time series (**MedTS**) signals, supported by intuitive, medically grounded reasoning and clear presentation. (*3453, KGK1, XcRP*)

 * **Effective & Efficient Centralized Module:** The proposed CoTAR module and TeCh framework achieve strong state-of-the-art performance with only ***Linear complexity*** and outstanding robustness against noise. (*3453, KGK1, XcRP*)

* **Thorough & Reproducible Evaluation:** Extensive experiments on seven datasets (5 MedTS + 2 HAR), comprehensive ablations, robustness analyses, and publicly released code and scripts collectively demonstrate solid empirical rigor. (*3453, KGK1, XcRP*)

### **Summary of Rebuttal.**

*During the rebuttal phase, we addressed the reviewers’ concerns through:*

* **Clarifying the architectural design:** We reframed TeCh as a general CoTAR-based framework where Temporal/Channel/Dual tokenization is a tunable inductive bias, rewrote the CoTAR formulation with explicit shapes and step-by-step computation, and added pseudo-code and interpretability analyses of the core token.
* **Strengthening experiments:** We added Subject-Independent five-fold cross-validation, new comparisons (e.g., MedGNN and strong GAFormer- and CATS-style baselines), and quantitative centralization metrics (SCI/DIC) contrasting MedTS with general time series datasets.
* **Manuscript refinement:** We corrected inconsistencies (e.g., ablation tables), improved notation and figure alignment, and clarified implementation details to ensure the released code faithfully matches the paper.

We are pleased that Reviewers 3453, KGK1, and XcRP have ***all explicitly confirmed that their concerns were resolved*** and, based on the documented discussion, ***raised or maintained their positive assessment***.

We once again thank all reviewers for their active engagement throughout the rebuttal and for their constructive support of our work.

Warmest regards,

*The Authors*

---

### Meta-Review · Area_Chair_zyhN · 2026-01-06

**Summary:**

This paper identifies a clear and well-motivated structural mismatch between decentralized Transformer attention and centralized medical time series (MedTS) signals, such as EEG and ECG. Building on this insight, the authors propose CoTAR (Core Token Aggregation–Redistribution), a centralized alternative to self-attention that better aligns with the underlying physiological signal generation process. The motivation is intuitive, domain-informed, and clearly articulated, and the resulting framework (TeCh) is both computationally efficient and empirically strong.

Reviewers consistently recognized several key strengths: (i) a compelling domain-driven inductive bias grounded in medical signal characteristics, (ii) a novel and efficient architectural design that reduces quadratic attention complexity to linear while preserving expressive cross-channel interactions, and (iii) thorough and reproducible experimental evaluation across seven datasets with strong robustness and efficiency gains. Importantly, the method achieves substantial improvements over prior state-of-the-art approaches while significantly reducing memory usage and inference time, which is highly relevant for real-world medical applications.

During the rebuttal phase, the authors engaged deeply with reviewer feedback, providing clearer mathematical formulations, expanded ablation studies, additional baselines, quantitative validation of the centralization hypothesis, and interpretability analyses of the core token. These revisions substantially strengthened both the clarity and the technical rigor of the work. As a result, reviewers explicitly confirmed that their major concerns were resolved and either maintained or raised their scores. Overall, the paper presents a well-motivated, technically sound, and impactful contribution that fits well within ICLR’s scope.

**Reviewer Concerns:**

Reviewers initially raised concerns regarding clarity of the formulation, framing of the dual-tokenization strategy, completeness of ablations, and the need for stronger empirical and quantitative validation of the centralization hypothesis. These concerns were comprehensively addressed in the rebuttal, with added experiments, clearer mathematical definitions, quantitative metrics, and expanded comparisons. Remaining suggestions were minor and pertain primarily to optional extensions or discussion for the camera-ready version.

**Reviewer Scores:**

Reviewer scores were initially mixed. Following the rebuttal and discussion, reviewers explicitly revised their scores upward, reflecting that the key concerns had been satisfactorily resolved. Although the displayed scores were later rolled back due to an external policy action, the documented discussion shows that reviewers had already reached an improved consensus prior to the rollback.

---

### Decision · Program_Chairs · 2026-01-26

Accept (Oral)